# Zoom-Zero: Reinforced Coarse-to-Fine Video Understanding via Temporal Zoom-in

## Abstract

Grounded video question answering (GVQA) aims to localize relevant temporal segments in videos and generate accurate answers to a given question; however, large video-language models (LVLMs) exhibit limited temporal awareness. Although existing approaches based on Group Relative Policy Optimization (GRPO) attempt to improve temporal grounding, they still struggle to faithfully ground their answers in the relevant video evidence, leading to temporal mislocalization and hallucinations. In this work, we present **Zoom-Zero**, a coarse-to-fine framework that first localizes query-relevant segments and then temporally zooms into the most salient frames for finer-grained visual verification. Our method addresses the limits of GRPO for the GVQA task with *two key innovations*: **(i)** a zoom-in accuracy reward that validates the fidelity of temporal grounding prediction and facilitates fine-grained visual verification on grounded frames; **(ii)** token-selective credit assignment, which attributes rewards to the tokens responsible for temporal localization or answer generation, mitigating GRPO's issue in handling multi-faceted reward signals. Our proposed method advances grounded video question answering, improving temporal grounding by 5.2% on NExT-GQA and 4.6% on ReXTime, while also enhancing average answer accuracy by 2.4%. Additionally, the coarse-to-fine zoom-in during inference further benefits long-form video understanding by preserving critical visual details without compromising global context, yielding an average improvement of 6.4% on long-video benchmarks. Our code will be publicly available[1].

## 1 Introduction

Large video-language models (LVLMs) have achieved remarkable progress in video understanding (Li et al., 2023b; 2024b; Cheng et al., 2024; Lin et al., 2024; Luo et al., 2023; Ataallah et al., 2024). However, current LVLMs often struggle to remain faithfully grounded in key visual evidence, leading to hallucinations when reasoning across video sequences. To evaluate this critical capability, video temporal grounding (VTG) (Gao et al., 2017; Anne Hendricks et al., 2017; Lei et al., 2021) measures how well models localize segments given an explicit event description, while the more comprehensive task of grounded video question answering (GVQA) (Xiao et al., 2024) requires models to implicitly infer the relevant moment from a general question for temporal localization and simultaneously generating accurate answers.

The key challenge of GVQA lies in achieving precise temporal localization while maintaining general video understanding capabilities. Reinforcement learning (RL) offers a promising solution for sharpening specific capabilities while preserving generalization from a pretrained LVLM (Lai et al., 2025). Recent efforts (Li et al., 2025b; Feng et al., 2025) have explored GRPO-based (Shao et al., 2024) RL algorithm for video temporal grounding and reasoning. However, most approaches (Wang et al., 2025b; Chen et al., 2025b) optimize with only format and Intersection over Union (IoU) rewards, neglecting the quality of the generated answers. Although VideoChat-R1 (Li et al., 2025b) incorporates an answer accuracy reward, these training objectives still cannot guarantee that localized video segments actually contain the visual evidence required for correct reasoning. Moreover, limited context budgets compel models to depend on coarse-grained representations, which overlook the

---

[1]Please refer to the anonymous GitHub link for access to the code.

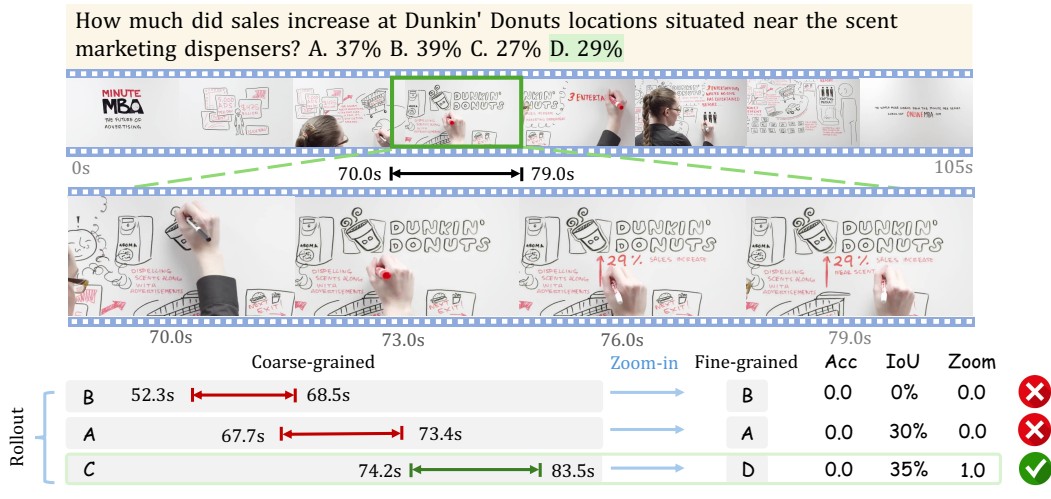

Figure 1: Our **Zoom-Zero** first rolls out samples to localize relevant segments with preliminary answers in the coarse-grained pass, then zooms into spotlight segments with higher-resolution video tokens in the fine pass. For example, the coarse pass may miss the small visual cue "29%", but the zoom-in captures the fine-grained details. This fine-grained visual verification (zoom-in accuracy reward) ensures that the temporally grounded segments truly provide key visual evidence.

fine-grained details critical for accurate question answering, a shortcoming that becomes especially severe in long videos where rich spatial information can be easily lost.

To address such challenges, we propose **Zoom-Zero** to achieve more accurate temporal grounding with finer-grained video understanding. First, we introduce *zoom-in accuracy* reward for GRPO in the GVQA task. It serves two critical roles: (1) verifying that grounded segments contain requisite evidence to answer the query, and (2) enabling dynamic context reallocation by zooming into key frames with increased spatial resolution. As illustrated in Figure 1, our approach first rolls out several samples to localize the relevant segments and produce preliminary answers in the coarse-grained pass. It then performs a fine-grained pass by narrowing down and zooming into the spotlight segments, dynamically allocating high-resolution video tokens. For instance, the coarse pass may overlook the small visual detail "29%" due to low-resolution tokens. Only by correctly grounding the segment and zooming into the relevant frames can the model capture such details and produce the right answer, thereby achieving the highest reward among all rollouts. This hierarchical paradigm resembles human visual cognition: breaking down complex problems, identifying relevant temporal intervals, and then refining focus to extract precise details.

In addition, when training with multi-faceted rewards (e.g., temporal localization accuracy, answer correctness) in the GVQA task, the standard GRPO algorithm (Shao et al., 2024) has key limitations. First, it compresses multiple reward signals into a single value via naïve summation, making it hard to differentiate targeted improvements for different aspects of the task. Second, the uniform credit assignment problem: it assigns an identical reward (advantage) to every token in a sequence based solely on the final outcome, regardless of weighting each token's contribution. We address this by introducing *token-selective credit assignment (TokenAdv)*, which selectively attributes credit to tokens specifically for temporal grounding or question answering, enabling finer-grained advantage estimation and more effective learning from diverse signals.

Extensive experiments demonstrate superior performance of Zoom-Zero across challenging benchmarks, including GVQA datasets NExT-GQA (Xiao et al., 2024), ReXTime (Chen et al., 2024a), and CG-Bench (Chen et al., 2025a), as well as long video understanding benchmarks VideoMME (Fu et al., 2025), MLVU (Zhou et al., 2025), and LVBench (Wang et al., 2024). Our method mutually enhances temporal grounding capability and question-answering performance. It advances temporal grounding by 5.2% on NExT-GQA and 4.6% on ReXTime. The main contributions of this work are:

- We introduce a zoom-in accuracy reward that verifies localized segments contain the visual evidence required for correct reasoning in a finer-grained manner, enhancing both localization precision and answer accuracy.

- We identify and address the limit of GRPO in handling multi-faceted reward signals by selective token-level credit assignment, enabling effective learning from diverse reward signals in GVQA.

- Our coarse-to-fine paradigm further enhances long-form video understanding by first coarsely identifying key segments and then zooming into fine-grained details, preserving global context while capturing critical information, resulting in an average 6.4% improvement on long-video benchmarks.

## 2 RELATED WORK

**Large Video Language Models.** Multimodal large language models (MLLMs) (Zhu et al., 2024; Liu et al., 2024b;a; Tong et al., 2024; Chen et al., 2023) have demonstrated remarkable progress in vision-language tasks. Recent advancements have further extended their capabilities to video understanding tasks (Li et al., 2023b; 2024b; Cheng et al., 2024; Lin et al., 2024; Luo et al., 2023; Ataallah et al., 2024). Large Video Language Models (LVLMs) process videos by extracting and encoding frames, and then rearranging them into final video representations. Some approaches (Li et al., 2023b; 2024b; Cheng et al., 2024) leverage the Q-Former module from BLIP-2 (Li et al., 2023a) to integrate visual and textual features, while others (Lin et al., 2024; Luo et al., 2023; Ataallah et al., 2024) directly concatenate frame features. To address intensive video tokens for long videos, several works train on sparsely sampled frames (Li et al., 2023b; Ataallah et al., 2024; Cheng et al., 2024; Zhang et al., 2024b; Li et al., 2024a), while others try to handle long videos by token pooling (Maaz et al., 2023; Li et al., 2024c; Song et al., 2024), token compression (Shen et al., 2025), memory aggregation (He et al., 2024), or frame selection (Hu et al., 2025; Zhang et al., 2025; Wu et al., 2019; Tang et al., 2025). Unlike frame-selection methods that search in the embedding space and select a fixed set of frames, our approach tackles the long-video token challenge by explicitly enhancing the model's temporal grounding capability through reasoning over the user query.

**Grounded Video Question Answering.** Video Temporal Grounding (Gao et al., 2017; Anne Hendricks et al., 2017; Lei et al., 2021) localizes relevant segments given an explicit event description. The more advanced task of Grounded Video Question Answering (GVQA) (Xiao et al., 2024) requires models to implicitly infer the relevant segment from a general question to perform localization and question-answering jointly. Recent LVLM-based approaches reformulate grounding as text generation (Nie et al., 2024; Ren et al., 2024; Huang et al., 2024a; Li et al., 2025b; Feng et al., 2025) while other methods (Wang et al., 2025a; Huang et al., 2024b) expand vocabularies to learn temporal embeddings for improved precision. Our approach leverages Qwen2.5VL (Bai et al., 2025) to predict textual temporal spans and introduces a novel coarse-to-fine training paradigm: initially predicting coarse timestamps for global localization, then dynamically zooming into identified segments for high-resolution visual verification. In contrast with a concurrent work (Li et al., 2025c) that relies on separate off-the-shelf VideoQA models to answer the query based on localized segments, our unified framework seamlessly integrates temporal grounding with question-answering within a single model for coherent video understanding.

**Reinforcement Learning for Grounded Video Question Answering.** Reinforcement learning (RL) has emerged as a powerful paradigm for improving the reasoning ability of large language models. Breakthroughs such as OpenAI-o1 (Jaech et al., 2024) and DeepSeek-R1 (Guo et al., 2025) have demonstrated notable success in addressing complex problems. DeepSeek-R1 (Guo et al., 2025) adopts group relative policy optimization (GRPO) to train LLMs to incentivize reasoning capability at inference time. Recently, RL has been adapted to LVLMs with the goal of strengthening video reasoning (Li et al., 2025b; Feng et al., 2025). Time-R1 (Wang et al., 2025b) and TVG-R1 (Chen et al., 2025b) adopt a two-stage pipeline, beginning with supervised fine-tuning (SFT) as a cold start, followed by GRPO-based RL training, while TimeZero (Wang et al., 2025b) demonstrates that a purely GRPO approach can be more effective without an SFT stage. These methods leverage only format and Intersection over Union (IoU) reward, whereas VideoChat-R1 (Li et al., 2025b) further integrates answer accuracy into RL training. In this work, we enhance GRPO by decoupling multi-faceted reward signals for selective token-level advantage estimation.

## 3 PRELIMINARY

**GRPO.** Group Relative Policy Optimization (GRPO) (Shao et al., 2024) is a variant of Proximal Policy Optimization (PPO) (Schulman et al., 2017) for reinforcement learning. Unlike PPO, which relies on a critic model, GRPO directly compares groups of candidate responses. This design eliminates the dependency on a critic, thereby substantially reducing training costs. Given a question-answer pair $(q, a)$, policy $\pi_{\theta_{old}}$ generates $G$ distinct candidate responses $o = o_1, \ldots, o_G$ through policy sampling. Then, the verifiable reward(s) $r_1, \ldots, r_G$ is calculated for each response. GRPO normalizes the scores by computing their mean and standard deviation, and then evaluates the relative quality of the responses accordingly.

$$A_{i,t} = \frac{r_i - \text{mean}(\{r_i\}_{i=1}^{G})}{\text{std}(\{r_i\}_{i=1}^{G})}, \tag{1}$$

where $A_{i,t}$ denotes the relative quality of the $t$-th token in $i$-th response. GRPO promotes higher-scoring answers within each group while regularizing the policy $\pi_\theta$ against the reference parameters $\pi_{ref}$ via a KL-divergence penalty $D_{KL}(\cdot|\cdot)$, leading to the final objective:

$$\max_{\pi_\theta} \mathbb{E}_{(q,a),\{o_i\}_{i=1}^{G} \sim \pi_{\theta_{old}}(\cdot|q)} \left[ \frac{1}{G} \sum_{i=1}^{G} \frac{1}{|o_i|} \sum_{t=1}^{|o_i|} \left( \frac{\pi_\theta(o_{i,t}|q, o_{i,<t})}{\pi_{\theta_{old}}(o_{i,t}|q, o_{i,<t})} \cdot A_{i,t} - \beta \, D_{KL}(\pi_\theta \,\|\, \pi_{ref}) \right) \right], \tag{2}$$

where $\beta$ is a regularization coefficient, preventing excessive deviation from the reference policy during optimization.

**Dynamic Spatiotemporal Resolution.** Qwen2.5-VL (Bai et al., 2025) dynamically adjusts tokens to determine the number of tokens per frame under a fixed video context budget. Specifically, the video context size is denoted as $L_v$, the maximum tokens per frame as $V_{max}$, the minimum tokens per frame as $V_{min}$, the video duration as $F$ seconds, and the sampling rate as $s$ frames per second. Based on these, the per-frame token resolution $V_{res}$ is defined as follows:

$$N = \min\left(F * s, \frac{L_v}{V_{min}}\right), \quad V_{res} = \max\left(V_{min}, \min(\frac{L_v}{N}, V_{max})\right) \tag{3}$$

## 4 ZOOM-ZERO

We propose a coarse-to-fine framework for grounded video question answering: a coarse-grained pass predicts query-conditioned intervals, followed by a fine-grained zoom-in that takes as input only the localized segments at higher per-frame token resolution (Section 4.1). Beyond standard format, IoU, and answer-accuracy rewards, we introduce a zoom-in accuracy reward to verify evidence within the localized span (Section 4.2). To overcome GRPO's limit in uniform credit assignment, we develop token-selective credit assignment for finer-grained advantage estimation tailored to multi-faceted rewards in the GVQA task (Section 4.3).

### 4.1 COARSE-TO-FINE VIDEO UNDERSTANDING VIA TEMPORAL ZOOM-IN

While dynamic token allocation offers flexibility, a fundamental trade-off remains: capturing long-range temporal context versus preserving fine-grained visual detail. Spatial or temporal downsampling inevitably discards critical information. This problem is exacerbated in longer videos, where preserving more frames often comes at the expense of per-frame spatial granularity. A coarse-to-fine strategy provides a principled remedy: a coarse pass preserves temporal context, followed by a fine-grained stage that processes evidence-bearing segments through temporal zoom-in[2].

More specifically, we leverage the model's temporal grounding capability to perform a fine-grained zoom-in on relevant segments and recover the details of the video. From the coarse view of the video,

---

[2]We term it temporal zoom-in, not spatiotemporal, to avoid confusion, since no spatial regions are predicted from the model. However, the spatio-temporal grid size per token is increased, since salient frames are sampled at higher temporal resolution (if the full video was sparsely sampled in the coarse pass), and spatial resolution is dynamically increased.

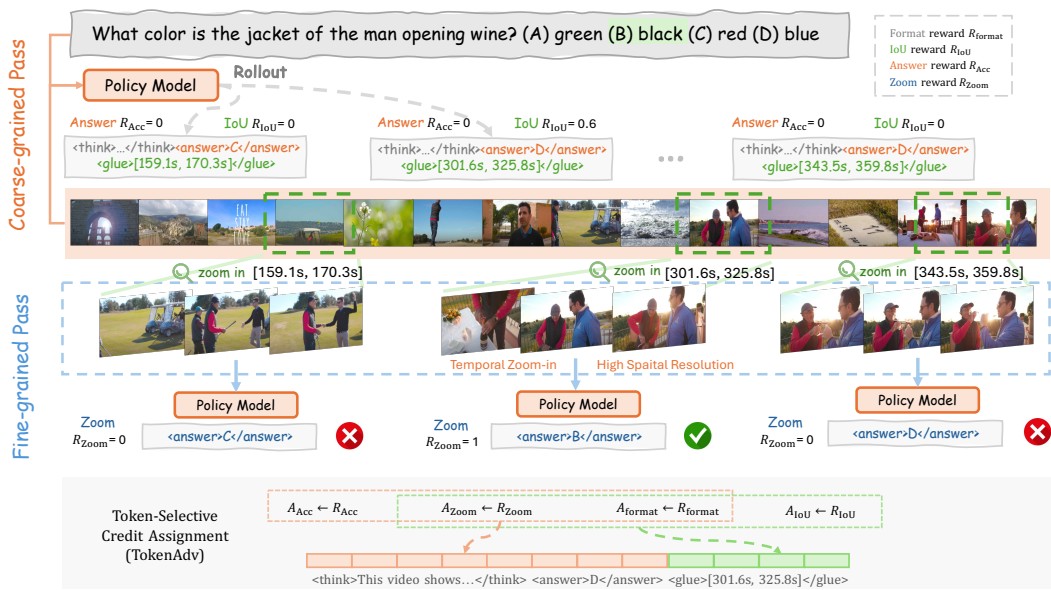

Figure 2: We present **Zoom-Zero**, a coarse-to-fine training pipeline that first rolls out samples to localize relevant segments with preliminary answers, followed by a fine-grained pass by zooming into spotlight segments and dynamically allocating high-resolution video tokens. The zoom reward enforces fine-grained visual verification of the predicted temporal span. In this example, only a faithful span prediction with the correct final answer yields the highest reward. Then we propose token-selective credit assignment (TokenAdv) for a finer-grained advantage estimation.

we obtain grounded start–end pairs $(s_1, e_1), (s_2, e_2), \ldots, (s_n, e_n)$. We crop the video accordingly, yielding $N' < N$ frames. Under a fixed visual token budget $L_v$, the per-frame video tokens increases from $V_{\text{res}} = \frac{L_v}{N}$ to $V'_{\text{res}} = \frac{L_v}{N'} > V_{\text{res}}$, enabling more fine-grained visual verification of the selected segments. This coarse-to-fine temporal zoom-in preserves global context while concentrating high-resolution capacity on the frames that matter most.

Crucially, this paradigm hinges on accurate, query-conditioned temporal grounding. To this end, we leverage GRPO-based reinforcement learning with carefully designed rewards that jointly improve temporal grounding and question answering, as detailed in the following section.

## 4.2 REWARDS DESIGN

In this section, we first review the basic rewards used in GVQA, i.e., format, temporal grounding, and answer accuracy, and then introduce our proposed zoom-in reward for fine-grained visual verification.

**Format Reward.** To guide the model toward producing responses in the desired format, we require the output to follow the instructions below:

---

**Format Prompt and Template**

Answer the question: `[QUESTION]` according to the content of the video. Select the answer from: `[OPTIONS]`. Output key information relevant to the question and options, marking precise timestamps or time ranges in seconds within `<time> </time>` tags, and present them in an interleaved analysis format. Enclose the full analysis in `<think> </think>` tags. Then, provide your answer within the `<answer> </answer>` tags, output the corresponding letter of the option. At the same time, in the `<glue> </glue>` tags, include only the precise video segments (in seconds) that strongly support your answer, in the format of [(s1, e1), (s2, e2), ...]. For example: `<answer>A</answer><glue>` [(20.3, 30.8)] `</glue>`.

---

We then apply regular expression matching to verify whether the model output conforms to this format. $R_{\text{format}}$ is assigned as 1 if the format fully matches the template.

**Answer Accuracy Reward.** We define reward $R_{\text{Acc}}$ to evaluate the correctness of the policy model's answer in coarse understanding by taking as input the whole video.

**Temporal Grounding Reward.** For temporal grounding, the model is required to predict a timestamp interval that specifies the video segment relevant to the given textual query. To evaluate this prediction, we adopt the Intersection over Union (IoU) between the model-predicted interval (from `<glue>` `</glue>`) and the ground-truth interval as the reward function. $R_{\text{IoU}} = \frac{|\mathcal{I}_{\text{pred}} \cap \mathcal{I}_{\text{gt}}|}{|\mathcal{I}_{\text{pred}} \cup \mathcal{I}_{\text{gt}}|}$, where $\mathcal{I}_{\text{pred}}$ and $\mathcal{I}_{\text{gt}}$ are the predicted time intervals and the ground truth intervals.

**Zoom Accuracy Reward.** Based on the temporal grounding prediction (from `<glue>` `</glue>`) in the coarse pass, we can obtain a set of salient frames that enables fine-grained visual verification. In the finer-grained pass, the model takes as input the question and the zoomed-in frames from the coarse response to produce the final answer. The reward $R_{\text{Zoom}}$ is assigned a value of 1 if the model produces an accurate final answer. This reward provides two key benefits: (1) enabling visual verification to ensure the predicted timestamp is accurately grounded in the relevant frames, and (2) facilitating a coarse-to-fine visual zoom-in to capture details within key frames.

### 4.3 TOKEN-SELECTIVE CREDIT ASSIGNMENT FOR ADVANTAGE ESTIMATION

Since our approach involves multiple rewards, i.e., $R_{\text{format}}$, $R_{\text{Acc}}$, $R_{\text{Zoom}}$, and $R_{\text{IoU}}$, the key question becomes how to leverage them for the policy updates. Standard GRPO handles multi-faceted rewards by naïvely summing them into a single scalar, thereby collapsing the contributions of individual reward signals. The advantage is then estimated only from this aggregated value (Equation 1), which cannot be decoupled for gradient updates. As a result, the model receives no explicit guidance on which aspect of its behavior each reward reflects, making it difficult to attribute feedback to specific abilities. In addition, the same advantage is assigned uniformly across all tokens in a response, which hides the contribution of each token from its corresponding rewards. Appendix C provides a simple example illustrating this limitation.

To overcome these limitations, we propose TokenAdv, a token-selective credit assignment for fine-grained token-level advantage estimation. Instead of summing up all rewards into one value for advantage estimation, we decouple advantage calculation separately for each reward type (Equation 4). In our case, since the outputs for answering and temporal grounding are explicitly formatted with task-specific tokens, it is feasible to distinguish the contribution of corresponding tokens to each aspect. Specifically, the token-level advantage is computed by averaging the relevant task-specific advantages for each token (Equation 5). This design allows the model to attribute feedback to specific rewards, improving its ability to learn from diverse, multi-faceted signals.

$$A_i^k = \frac{r_i^k - \text{mean}(\{r_i^k\}_{i=1}^G)}{\text{std}(\{r_i^k\}_{i=1}^G)}, \quad r^k \in \{R_{\text{format}}, R_{\text{Acc}}, R_{\text{Zoom}}, R_{\text{IoU}}\} \tag{4}$$

$$A_{i,t} = \begin{cases} \text{mean}(A_i^{\text{format}}, A_i^{\text{Zoom}}, A_i^{\text{IoU}}) & \text{if } o_{i,t} \in \text{<glue>} \cdots \text{</glue>} \\ \text{mean}(A_i^{\text{format}}, A_i^{\text{Zoom}}, A_i^{\text{Acc}}) & \text{else} \end{cases} \tag{5}$$

By selectively assigning credits to task-specific tokens, we guide policy gradient updates toward the most influential parts of the output for each capability. This targeted credit assignment allows the model to effectively leverage diverse reward signals, leading to improved temporal grounding and question-answering performance, as shown in Figure 4.

## 5 EXPERIMENTS

### 5.1 EXPERIMENTAL SETUPS

**Benchmarks and Evaluation Metrics.** We evaluate on three GVQA benchmarks: NExT-GQA (Xiao et al., 2024), ReXTime (Chen et al., 2024a), and CG-Bench (Chen et al., 2025a). Temporal grounding is measured by mean Intersection-over-Union (mIoU), R@0.3 (IoU > 0.3), and R@0.5 (IoU > 0.5); video understanding by multiple-choice question (MCQ) accuracy; Acc@GQA measures the percentages of questions that are correctly answered and visually grounded, i.e., IoP≥ 0.5, where

Table 1: Grounded video question answering results on NExT-GQA (Xiao et al., 2024) and ReX-Time (Chen et al., 2024a). All models are of comparable scale (7B or 8B).

| Models | NExT-GQA | | | | ReXTime | | | |
|---|---|---|---|---|---|---|---|---|
| | Acc@GQA | mIoU | R@0.3 | R@0.5 | Acc | mIoU | R@0.3 | R@0.5 |
| Qwen2.5-VL (Bai et al., 2025) | 18.9 | 20.2 | 31.6 | 18.1 | 51.1 | 27.4 | 36.1 | 24.8 |
| *SFT-based* | | | | | | | | |
| TimeChat (Ren et al., 2024) | 7.6 | 20.6 | 34.1 | 17.9 | 40.0 | 11.6 | 14.4 | 7.6 |
| VTimeLLM (Huang et al., 2024a) | 17.4 | 24.4 | 36.1 | 20.1 | 36.1 | 20.1 | 28.8 | 17.4 |
| Grounded-VideoLLM (Wang et al., 2025a) | 26.7 | 21.1 | - | 18.0 | - | - | - | - |
| *RL-based* | | | | | | | | |
| VideoChat-TPO (Li et al., 2025a) | 25.5 | 27.7 | 41.2 | 23.4 | - | 25.2 | 34.5 | 19.3 |
| TVG-R1 (Chen et al., 2025b) | 22.1 | 29.2 | 41.6 | 20.8 | 53.6 | 28.2 | 41.0 | 24.5 |
| VideoChat-R1 (Li et al., 2025b) | 24.3 | 32.4 | 50.2 | 27.7 | 58.1 | 38.6 | 50.6 | 39.0 |
| Zoom-Zero (Ours) | **29.0** | **37.6** | **55.6** | **33.8** | **62.0** | **43.2** | **56.5** | **44.1** |

IoP is the intersection over prediction span. CG-Bench (Chen et al., 2025a), a long-form GQA benchmark, additionally introduces two metrics rec.@IoU and acc.@IoU: rec.@IoU averages recall over IoU thresholds $\{0.1, 0.2, 0.3, 0.4, 0.5\}$ to estimate the probability of correctly retrieving clue intervals; acc.@IoU counts a response as correct only if the predicted answer is accurate and its IoU exceeds the threshold, and is averaged over the same thresholds per the original protocol. We also assess general video understanding on four long-video benchmarks, VideoMME (Fu et al., 2025), MLVU (Zhou et al., 2025), LVBench (Wang et al., 2024) and CG-Bench (Chen et al., 2025a), and report MCQ accuracy.

**Baselines.** We compare our approach with several strong baselines, including SFT-based LVLMs with grounding capability such as VTimeLLM (Huang et al., 2024a), TimeChat (Ren et al., 2024) and Grounded-VideoLLM (Wang et al., 2025a), RL-based methods VideoChat-TPO (Li et al., 2025a), TVG-R1 (Chen et al., 2025b) VideoChat-R1 (Li et al., 2025b) as well as general-purpose LVLMs such as LLaVA-OneVision (Li et al., 2024a), Qwen2.5-VL (Bai et al., 2025) and InternVL2.5 (Chen et al., 2024b). All open-sourced models are of comparable scale (7B or 8B). All SFT-based models are evaluated in a zero-shot setting on NExT-GQA (Xiao et al., 2024). RL methods, VideoChat-R1 (Li et al., 2025b), and our model are trained on the NExT-GQA val split. For ReXTime (Chen et al., 2024a) (Table 1, right) and CG-Bench (Chen et al., 2025a) (Table 2, rightmost), all models are evaluated strictly in the zero-shot setting, ensuring a valid and fair comparison across methods.

**Training Details.** We adopt Qwen2.5-VL-7B (Bai et al., 2025) as the base model. The maximum number of video tokens is set to 8192, with videos sampled at 1 fps during training. The minimum video frame resolution is $16 \times 28 \times 28$ pixels and the maximum is $768 \times 28 \times 28$, allowing the number of tokens per frame to be adaptively adjusted under the video context budget. The maximum response length is capped at 512 tokens. The statistics of training data are shown in Appendix A and Table 7. All experiments are performed on NVIDIA A100 GPUs (80GB), with a global batch size of 64. Further implementation details are provided in the Appendix B.

## 5.2 MAIN RESULTS

**Grounded Video Question Answering.** We evaluate grounded video question answering on NExT-GQA (Xiao et al., 2024) and ReXTime (Chen et al., 2024a), reporting both answer accuracy and temporal grounding quality as shown in Table 1. Our model achieves state-of-the-art performance across all metrics on both benchmarks, surpassing strong RL-based baselines such as VideoChat-R1 (Li et al., 2025b). Notably, on NExT-GQA (Xiao et al., 2024), we improve mIoU by 5.2%, R@0.3 by 5.4%, and R@0.5 by 6.1% over the runner-up. On ReXTime (Chen et al., 2024a), our model also consistently yields an average improvement of 4.6% across all metrics.

We also report GQA performance on the challenging CG-Bench (Chen et al., 2025a) in Table 2, which contains very long videos where the answer-supporting clue typically occupies $\leq 1\%$ of the total duration. Beyond IoU, we evaluate how well the predicted segment covers the ground-truth

Table 2: Performance on long video understanding (VideoMME (Fu et al., 2025), MLVU (Zhou et al., 2025), LVBench (Wang et al., 2024)) and long GVQA (CG-Bench (Chen et al., 2025a)) tasks. All open-sourced models are of comparable scale (7B or 8B).

| Models | VideoMME (*w/o & w/ sub.*) | | MLVU | LVBench | CG-Bench | | |
|---|---|---|---|---|---|---|---|
| | Overall | Long | M-Avg | Avg | mIoU | rec.@IoU | acc.@IoU |
| Duration | 1010s | 2386s | 651s | 4101s | | 1624s | |
| *Proprietary LVLMs* | | | | | | | |
| Gemini 1.5 Pro (Google, 2024) | 75.0 / 81.3 | 67.4 / 77.4 | - | 33.1 | 3.85 | 5.61 | 2.64 |
| GPT-4o (OpenAI, 2024) | 65.3 / 77.2 | 65.3 / 72.1 | 64.6 | 30.8 | 5.73 | 8.12 | 4.33 |
| *Open-Source LVLMs* | | | | | | | |
| LLaVA-OneVision (Li et al., 2024a) | 58.2 / 61.5 | - / - | 64.7 | - | 1.56 | 1.19 | 1.72 |
| LongVA (Zhang et al., 2024a) | 52.6 / - | 46.2 / - | 56.3 | - | 2.91 | 3.15 | 1.32 |
| InternVL2.5 (Chen et al., 2024b) | 64.2 / 66.9 | - / - | 68.9 | 38.4 | - | - | - |
| Qwen2.5-VL (Bai et al., 2025) | 65.2 / 70.7 | 51.1 / 62.0 | 70.2 | 45.3 | 2.48 | 3.15 | 1.36 |
| TVG-R1 (Chen et al., 2025b) | 64.3 / 69.1 | 52.7 / 62.4 | 69.7 | 42.3 | 2.43 | 3.62 | 1.29 |
| VideoChat-R1 (Li et al., 2025b) | 64.3 / 69.1 | 53.4 / 62.3 | 69.5 | 43.7 | 5.91 | 8.38 | 2.56 |
| Zoom-Zero (Ours) | **66.0 / 71.2** | **54.8 / 64.2** | **70.8** | **45.7** | **6.68** | **9.30** | **3.62** |

clue span using Intersection-over-Ground truth (IoG; see Appendix D.1). As shown in Table 3, our model achieves a 7.7% gain of mIoG over the runner-up, validating that the zoom-in accuracy reward $R_{\text{Zoom}}$ encourages predictions that not only localize the relevant temporal segments but also better cover the most salient frames containing key visual cues.

Table 3: Temporal grounding coverage ratio. For ReXTime (Chen et al., 2024a), results (IoU and accuracy) are only obtainable via server submission without access to ground-truth spans; therefore, to report mIoG (mean Intersection-over-Ground truth), we use the validation set for comparison.

| Models | NExT-GQA | | | ReXTime val | | | CG-Bench | | |
|---|---|---|---|---|---|---|---|---|---|
| | mIoU | mIoG | mIoP | mIoU | mIoG | mIoP | mIoU | mIoG | mIoP |
| Qwen2.5-VL (Bai et al., 2025) | 20.2 | 56.8 | 29.5 | 31.6 | 54.3 | 43.2 | 2.48 | 10.35 | 4.16 |
| VideoChat-R1 (Li et al., 2025b) | 32.4 | 93.5 | 39.1 | 43.5 | 64.3 | 52.8 | 5.91 | 18.44 | 7.34 |
| Zoom-Zero (Ours) | **37.6** | **94.7** | **43.2** | **44.7** | **67.6** | **53.5** | **6.68** | **26.15** | **8.25** |

**Long Video Understanding.** We compare against general-purpose LVLMs with temporal grounding capability and RL-based models explicitly optimized for grounding (TVG-R1 (Chen et al., 2025b), VideoChat-R1 (Li et al., 2025b)). While RL approaches that prioritize grounding can trade off general GQA accuracy, our method proposes token-selective credit assignment that decouples reward signals from answer accuracy and temporal grounding, assigning credit to the appropriate tokens. This mitigates the accuracy–grounding trade-off and yields stronger temporal localization without degrading question answering, as shown in Table 2.

**Qualitative Results.** We provide qualitative results in Figures 5 to 8 in the Appendix to demonstrate the model's performance on the GVQA task. For example, as shown in Figure 5, the model can localize each event mentioned in the question and arrange them in the correct chronological order.

Table 4: Long video understanding via temporal zoom-in evaluated with MCQ accuracy.

| Models | VideoMME (*long w/ sub.*) | MLVU | LVBench | CG-Bench |
|---|---|---|---|---|
| Qwen2.5-VL (Bai et al., 2025) | 62.0 | 70.2 | 45.3 | 29.2 |
| Zoom-Zero (Ours) | 64.2 | 70.8 | 45.7 | 36.1 |
| Zoom-Zero + Coarse-to-fine (Ours) | 66.2 | 71.4 | 46.3 | 39.0 |
| Zoom-Zero + Divide-and-conquer (Ours) | **68.7** | **73.4** | **48.1** | **42.2** |

## 5.3 Long Video Understanding via Temporal Zoom-in

The above experiments demonstrate our model's ability to answer questions while faithfully localizing relevant video segments. Although our primary goal is to enhance GVQA, we further present two strategies that further benefit long-video understanding through temporal zoom-in.

**Coarse-to-Fine.** In long-video scenarios, we first let the model trade spatial resolution for broad temporal coverage to obtain a global overview. Once it has a coarse understanding and localizes the query-relevant interval, we enable a fine-grained pass at higher spatial resolution for frames of interest as mentioned in Section 4.1. As Table 4 (Coarse-to-fine) shows, it consistently improves performance by providing targeted visual verification on a small set of salient frames with higher spatial resolution, thus enhancing fine-grained understanding. We provide spatial and temporal resolution before and after zoom-in in Table 13 and qualitative results in Figures 9 to 11 in the Appendix.

**Divide-and-Conquer.** Another strategy is to partition a long video into non-overlapping windows and perform a temporal search over them. For each window, the model predicts a query-relevant temporal span and an answer. We then aggregate frames from spans with high-confidence answers and apply a fine-grained zoom-in. Answer confidence is computed as the probability of the predicted answer token, where $c = p_{\pi_\theta}(t)$ for token $t$ strictly between `<answer>` and `</answer>`. We select the top spans based on answer confidence and aggregate those frames as input to obtain the final answer. As shown in Table 4 (Divide-and-conquer), it yields an average +6.4% improvement over the baseline Qwen2.5-VL (Bai et al., 2025). Please refer to Appendix D.3 for ablation studies on the window size and the number of aggregated predicted temporal spans.

Table 5: Ablation studies.

| Models | NExT-GQA | | | | ReXTime | | | |
|---|---|---|---|---|---|---|---|---|
| | Acc | mIoU | R@0.3 | R@0.5 | Acc | mIoU | R@0.3 | R@0.5 |
| Qwen2.5-VL (Bai et al., 2025) | 53.3 | 20.2 | 31.6 | 18.1 | 51.1 | 27.4 | 36.1 | 24.8 |
| + GRPO ($R_{\text{format}}+R_{\text{IoU}}+R_{\text{Acc}}$) | 69.6 | 35.3 | 52.5 | 29.9 | 58.3 | 39.2 | 51.8 | 39.6 |
| + GRPO + TokenAdv | 69.9 | 36.9 | 54.9 | 32.3 | 59.8 | 41.5 | 53.9 | 41.7 |
| + GRPO + $R_{\text{Zoom}}$ | 70.4 | 36.3 | 53.8 | 31.4 | 60.2 | 40.9 | 53.0 | 40.9 |
| + GRPO + $R_{\text{Zoom}}$ + TokenAdv | **70.7** | **37.6** | **55.6** | **33.8** | **62.0** | **43.2** | **56.5** | **44.1** |

## 5.4 Ablation Studies

**Impact of Each Component.** As shown in Table 5, introducing TokenAdv improves grounding performance over baseline GRPO, i.e., NExT-GQA mIoU 35.3→36.9; ReXTime mIoU 39.2→41.5. The zoom-in reward further boosts answer quality and grounding with larger gains in accuracy (+1.9) on ReXTime over GRPO. Combining both components yields the best performance across all metrics: NExT-GQA accuracy (+1.1) and mIoU (+2.3); ReXTime accuracy (+3.7) and mIoU (+4.0), which proves that the selective credit assignment and zoom-in verification enhance both temporal localization and evidence-faithful answering.

**Duration Analysis.** Figure 3 in the Appendix shows the grounding accuracy upon clue duration/portion on NExT-GQA. We observe that shorter ground-truth spans or smaller clue portions make temporal grounding more challenging. Nevertheless, our method consistently improves over the base GRPO across all ranges, demonstrating stronger robustness to temporal variations.

## 5.5 Computation Analysis

**Training computation.** By using 8×A100 GPUs with a global batch size of 64, we measure the average per-step training time and TFLOPs. Without the temporal zoom-in paradigm, each step takes 13.29 minutes and consumes 551.2 TFLOPs. Incorporating the temporal zoom-in strategy increases the per-step time to 15.30 minutes and TFLOPs to 585.4, representing a 1.15× increase in training time per step.

Table 6: Trade-off between inference speed and accuracy gain. *Time* denotes the average inference time per video.

| Models | MLVU | | LVBench | | VideoMME *long (w/ sub.)* | |
|---|---|---|---|---|---|---|
| | Acc | Time | Acc | Time | Acc | Time |
| Duration | 651s | | 4101s | | 2386s | |
| Qwen2.5-VL (Bai et al., 2025) | 70.2 | 18.5s | 45.3 | 39.7s | 62.0 | 25.6s |
| Zoom-Zero (Ours) | 70.8 | 18.7s | 45.7 | 40.6s | 64.2 | 25.8s |
| Zoom-Zero + Coarse-to-fine (Ours) | 71.4 | 31.1s | 46.3 | 55.5s | 66.2 | 35.2s |
| Zoom-Zero + Divide-and-conquer (Ours) | 73.4 | 33.5s | 48.1 | 110.5s | 68.7 | 59.7s |

**Inference speed analysis.** We provide a clearer breakdown of the effectiveness–latency trade-off in the table below, and report three inference scenarios in Table 6

(i) **One-stage inference:** Zoom-Zero (second row) uses the same one-stage inference pipeline as the baseline, resulting in nearly identical inference time (it might vary a little due to the number of generated tokens). Trained with our proposed method, this setting yields an average improvement of +1.0 over the baseline without introducing additional latency. Please kindly note that the main experimental results as shown in Table 1 and Table 2 only have one-stage inference. (ii) **Two-stage inference (Coarse-to-fine):** The coarse-to-fine variant adds a fine-grained pass on grounded frames. This introduces a moderate increase in computation, approximately 1.4× inference time, while delivering a higher average absolute improvement of +2.1 over the baseline. (iii) **Two-stage inference (Divide-and-conquer):** The divide-and-conquer scheme is an optional test-time scaling strategy designed to further push performance. While it increases inference time to around 2.3×, it also achieves the largest gain, improving the baseline by +4.3 on average.

# 6 CONCLUSION

We introduce Zoom-Zero, a coarse-to-fine framework for grounded video question answering that first localizes query-relevant segments, then zooms into salient frames to capture fine-grained details. Our approach enhances GRPO for GVQA with two key contributions: (i) a zoom-in accuracy reward for evidence-faithful temporal grounding and fine-grained visual verification, and (ii) token-selective credit assignment for advantage estimation, assigning credit to the tokens responsible for localization or answer generation, respectively, addressing GRPO's limits under multi-faceted reward signals. Our method improves both temporal grounding and answer accuracy, raising temporal grounding by 5.2% on NExT-GQA and 4.6% on ReXTime. Its coarse-to-fine paradigm boosts long-form video understanding by an average of 6.4%, preserving critical detail without sacrificing global context.

## ETHICS STATEMENT

Our work builds on large video-language models (LVLMs) and reinforcement learning for grounded video question answering. We do not collect or annotate any human subject data; all experiments use publicly available datasets under research licenses. We adhere to the terms of use specified by the original dataset creators and provide appropriate citations. Our approach does not introduce additional risks of data misuse or privacy leakage.

## REPRODUCIBILITY STATEMENT

We make every effort to ensure reproducibility of our results. Full implementation details are provided in Appendix B. All datasets used in our experiments are publicly accessible and described in Appendix A. We provide the evaluation protocols and metrics in Section 5.1, and present ablation studies to analyze the effect of key components in Section 5.4 and Appendix D.

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

# APPENDIX

## A    TRAINING DATA

Table 7 summarizes the statistics of the training datasets. For the QVHighlights (Lei et al., 2021) training split, which contains 9,996 examples, we only keep videos longer than 120 seconds. For the PLM-Video (Cho et al., 2025) multiple-choice split, we perform a quality check to remove examples that cannot be correctly answered using the full video, but can be correctly answered when restricted to the cropped segment defined by the clue duration. This ensures that the clue duration indeed provides sufficient information for identifying the correct video segment.

After Stage I training, we employ the model trained from Stage I for offline data filtering. Specifically, we generate $n = 8$ responses per example and discard those without meaningful reward signals. For question-answering ability, we exclude examples for which all generated responses answer the question correctly, as they lack discriminative signals. For temporal grounding ability, we filter out examples with low response variance. In particular, we retain only examples where the responses yield a sufficiently strong relative reward signal, quantified by the difference between the maximum IoU and the mean IoU across responses:

$$\delta = \max_{1 \leq i \leq n} \text{IoU}_i \ - \ \frac{1}{n}\sum_{i=1}^{n} \text{IoU}_i \tag{6}$$

We filter out examples with $\delta < 0.1$.

Table 7: **Statistics of training data.** NExT-GQA and ActivityNet in seconds stage are sampled from the first stage by filtering reward variation based on the first-stage model.

|          | Dataset                              | #Queries | Video Len. | Moment Len. |
|----------|--------------------------------------|----------|------------|-------------|
| Stage I  | NExT-GQA (Xiao et al., 2024)         | 3,358    | 43.9s      | 8.5s        |
|          | ActivityNet (Krishna et al., 2017)   | 4,727    | 177.3s     | 48.35       |
|          | QVHighlights (Lei et al., 2021)      | 7,218    | 150.0s     | 34.1s       |
| Stage II | ActivityNet (Krishna et al., 2017)   | 1,395    | 220.87     | 88.4s       |
|          | NExT-GQA (Xiao et al., 2024)         | 1,004    | 50.2s      | 7.1s        |
|          | PLM-Video (Cho et al., 2025)         | 5,333    | 808.6s     | 26.1s       |

## B    IMPLEMENTATION DETAILS

We adopt Qwen2.5-VL-7B (Bai et al., 2025) as the base model. The maximum number of video tokens is set to 8,192, with videos sampled at 1 fps during training. The minimum video frame resolution is $16 \times 28 \times 28$ pixels, and the maximum is $768 \times 28 \times 28$, allowing the number of tokens per frame to be adaptively adjusted under the video context budget. The maximum response length is capped at 512 tokens. Due to computational resource limitations, we conduct RL training in two stages. In the first stage, we train on 20K short-video GQA examples from NExT-GQA (Xiao et al., 2024), ActivityNet (Krishna et al., 2017), and QVHighlights (Lei et al., 2021). In the second stage, we train on the *yt1b_mcqa* split from PLM-Video (Cho et al., 2025), combined with the short-video data sampled from the first stage, for a total of 7K examples. The statistics of training data are shown in Appendix A and Table 7. All experiments are performed on NVIDIA A100 GPUs (80GB), with a global batch size of 64.

During inference, we evaluate all models at 1 FPS with a context size of 8,192 on the short-video benchmarks NExT-GQA (Xiao et al., 2024) and ReXTime (Chen et al., 2024a). For long-video benchmarks: CG-Bench (Chen et al., 2025a), VideoMME (Fu et al., 2025), MLVU (Zhou et al., 2025), and LVBench (Wang et al., 2024). We also uniformly sampled a maximum of 256 frames and set the context size to 16,384.

Table 8: A simple example to demonstrate the GRPO's uniform credit assignment problem.

| Response | $R_{\mathrm{IoU}}$ | $R_{\mathrm{Acc}}$ | $A_{\mathrm{IoU}}$ | $A_{\mathrm{Acc}}$ | $R_{\mathrm{Sum}}$ | $A_{\mathrm{Sum}}$ |
|---|---|---|---|---|---|---|
| 1 | 0.0 | 1.0 | -1.40 | +0.82 | 1.0 | +0.06 |
| 2 | 0.5 | 0.0 | +0.44 | -1.22 | 0.5 | -1.54 |
| 3 | 0.4 | 1.0 | +0.07 | +0.82 | 1.4 | +1.34 |
| 4 | 0.8 | 0.0 | +1.55 | -1.22 | 0.8 | -0.58 |
| 5 | 0.2 | 1.0 | -0.66 | +0.82 | 1.2 | +0.70 |

Table 9: Ablation on the number of generated responses $G$ per prompt during GRPO training.

| G | NExT-GQA | | | | RexTime | | | |
|---|---|---|---|---|---|---|---|---|
| | Acc | mIoU | R@0.3 | R@0.5 | Acc | mIoU | R@0.3 | R@0.5 |
| 2 | 69.6 | 33.7 | 50.3 | 27.5 | 58.5 | 36.9 | 49.0 | 37.1 |
| 4 | 69.8 | 35.2 | 52.6 | 29.6 | 58.8 | 40.1 | 53.2 | 40.6 |
| 8 | 70.7 | 37.6 | 55.6 | 33.8 | 62.0 | 43.2 | 56.5 | 44.1 |

## C LIMITATION OF GRPO IN UNIFORM CREDIT ASSIGNMENT

In Table 8, we present a simple example with two rewards, $R_{\mathrm{IoU}}$ and $R_{\mathrm{Acc}}$, across five responses to illustrate the limitations of GRPO arising from naïve reward summation and uniform credit assignment. For example, the first response attains the lowest temporal grounding reward, $R_{\mathrm{IoU}}^{(1)} = 0$, yet its overall advantage under standard GRPO is positive, $A_{\mathrm{Sum}}^{(1)} = +0.06$. In contrast, response 4 achieves much better temporal grounding, $R_{\mathrm{IoU}}^{(4)} = 0.8$, but receives a lower advantage, $A_{\mathrm{Sum}}^{(4)} = -0.58$. Due to uniform credit assignment, all tokens in response 1 are reinforced by the positive advantage, while all tokens in response 4 are penalized. This hides the contribution of tokens that support more accurate temporal grounding.

In contrast, computing separate advantages for each reward, $A_{\mathrm{IoU}}$ and $A_{\mathrm{Acc}}$, provides a clearer view of each task's contribution. By selectively assigning these decoupled advantages to the corresponding tokens, our approach TokenAdv, updates the policy to increase the probability of tokens that positively impact their respective tasks.

## D EXPERIMENTS

### D.1 TEMPORAL GROUNDING COVERAGE

In addition to IoU, we evaluate how well the predicted segment covers the ground-truth clue span using Intersection-over-Ground truth (IoG), defined as $\mathrm{IoG} = \frac{|\mathcal{I}_{\mathrm{pred}} \cap \mathcal{I}_{\mathrm{gt}}|}{|\mathcal{I}_{\mathrm{gt}}|}$, where $\mathcal{I}_{\mathrm{pred}}$ is the predicted temporal span and $\mathcal{I}_{\mathrm{gt}}$ is the ground-truth clue span. We report mean IoG (mIoG) as the average IoG across instances. IoG directly measures coverage of the ground truth and thus verifies whether temporal grounding captures the key frames relevant to the query, particularly informative for the finer-grained zoom-in.

We present results in Table 3. For ReXTime (Chen et al., 2024a), only IoU and accuracy are available via server-side evaluation, and the ground-truth clue spans are not released; consequently, we compute and report mIoG on the validation set for comparison. Our model improves mIoG by 1.2% on NExT-GQA (Xiao et al., 2024) and by 3.3% on ReXTime (Chen et al., 2024a). For very long videos such as CG-Bench (Chen et al., 2025a), mIoU can be less informative because the larger denominator depresses scores. Considering both mIoU and mIoG shows that our model not only localizes the relevant moments but also achieves strong coverage of key frames.

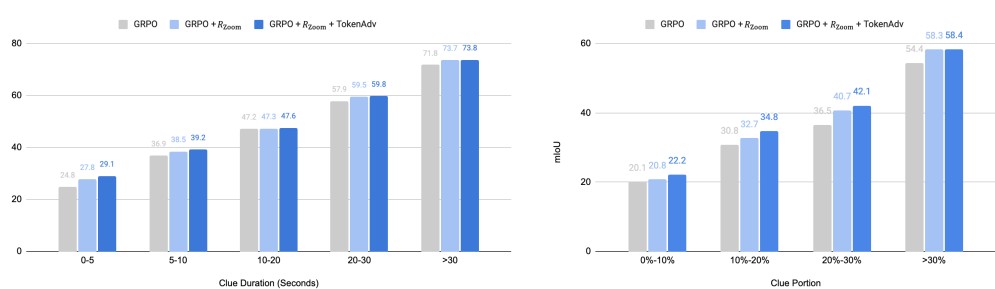

Figure 3: **Temporal grounding robustness analysis on NExT-GQA.** Left: mIoU results across different ground-truth clue durations. Right: mIoU results across different clue proportions (ground-truth clue duration relative to the total video duration).

## D.2   THE NUMBER OF GENERATED RESPONSES

We investigate the impact of the number of generated responses $G$ per prompt during GRPO training, as this hyperparameter directly influences the diversity and quality of the policy optimization process. As presented in Table 9, increasing $G$ from 2 to 8 consistently improves performance across both datasets and all evaluation metrics. Based on these results, we adopt $G = 8$ for all main experiments, as it provides the optimal balance between computational efficiency and performance gains.

## D.3   DIVIDE-AND-CONQUER

We study the impact of window size (Table 10) and the number of predicted temporal spans aggregated in the divide-and-conquer strategy. Because this approach requires scanning every sliding window during the coarse-grained pass, it introduces an average $\times 2.3$ increase in inference cost. Nevertheless, it improves performance across all long-video benchmarks by an average of +6.4%, demonstrating that our temporal zoom-in with higher spatial resolution provides substantial benefits for long video understanding. (Table 11) shows the impact of number of aggregated temporal spans with top answer confidence.

Table 10: Window size ablation.

| Window Size | VideoMME (*long w sub.*) | MLVU | LVBench |
| --- | --- | --- | --- |
| 128 | 67.4 | 72.1 | 47.6 |
| 256 | 68.7 | 73.4 | 48.1 |
| 384 | 68.4 | 72.4 | 48.5 |

Table 11: Number of aggregated temporal spans with top answer confidence.

| # Aggregated Spans | VideoMME (*long w sub.*) | MLVU | LVBench |
| --- | --- | --- | --- |
| 3 | 73.2 | 66.8 | 47.5 |
| 4 | 73.4 | 68.7 | 48.1 |
| 5 | 73.2 | 68.4 | 47.9 |

## D.4   COARSE-TO-FINE VIDEO UNDERSTANDING ON GVQA

We further evaluate short-form GVQA answer accuracy on NExT-GQA and ReXTime through temporal zoom-in, as reported in Table 12. Both benchmarks consist of short videos, where the model

Table 12: Grounded question answering (GQA) results on NExT-GQA (Xiao et al., 2024) and ReXTime (Chen et al., 2024a) with temporal zoom-in.

| Models | NExT-GQA | | | | ReXTime | | | |
|---|---|---|---|---|---|---|---|---|
| | Acc | mIoU | R@0.3 | R@0.5 | Acc | mIoU | R@0.3 | R@0.5 |
| Zoom-Zero | 70.7 | 37.6 | 55.6 | 33.8 | 62.0 | 43.2 | 56.5 | 44.1 |
| Zoom-Zero + Coarse-to-fine | 71.4 | N/A | N/A | N/A | 62.8 | N/A | N/A | N/A |

Table 13: Spatial and temporal resolution in coarse-to-fine video understanding.

| Benchmark | VideoMME Long | | MLVU | | LVBench | |
|---|---|---|---|---|---|---|
| Duration | 2386s | | 651s | | 4101s | |
| | Coarse | Fine | Coarse | Fine | Coarse | Fine |
| Avg frames | 256 | 136 | 253 | 86 | 256 | 154 |
| Avg FPS | 0.1 | 1.0 | 0.2 | 1.0 | 0.06 | 1.0 |
| Avg tokens/frame | 54 | 76 | 64 | 190 | 64 | 62 |

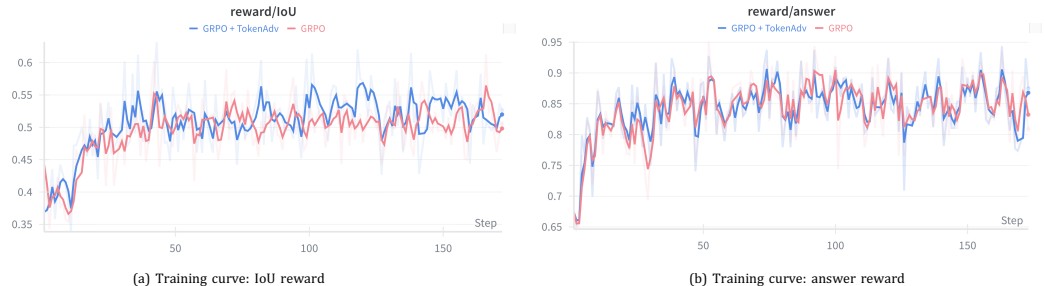

(a) Training curve: IoU reward       (b) Training curve: answer reward

Figure 4: Training curve: IoU reward $R_{IoU}$ and answer reward $R_{Acc}$ comparison with baseline GRPO and our improved GRPO with TokenAdv.

**Question:** Arrange the following events from the video in the correct chronological order: (1)The action of the lures is shown underwater as several different fish go after the lures; (2)Several men show off the different lures they are using for ice fishing; (3)The video ends with the closing credits and Graphics shown on the screen; (4)An introduction comes onto the screen for a video about fishing lures.
**Options:** (A) 3->2->1->4 (B) 4->2->1->3 (C) 1->2->3->4 (D) 2->1->3->4

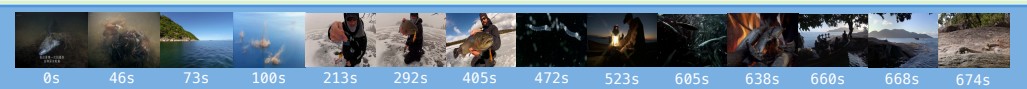

**Output:** <think>First, the video introduces the topic of fishing lures with an opening screen <time> (20.3, 49.7) </time>. Next, several men showcase the different lures they are using for ice fishing <time> (283.5, 304.2) </time>. Then, the action of the lures is demonstrated underwater as various fish chase after them <time> (593.1, 611.4) </time>. Finally, the video concludes with closing credits and graphics <time> (673.8, 680.2) </time>.</think>\n<answer>B</answer><glue>[(20.3, 49.7), (283.5, 304.2), (593.1, 611.4), (673.8, 680.2)]</glue>

Figure 5: A qualitative example for long video understanding.

can preserve most temporal context and details within the context budget. In this setting, the zoom-in paradigm improves performance by 0.7% on NExT-GQA and 0.8% on ReXTime.

**Question:** Are there any irregularities in this surveillance video? If there are, what sort are they?
**Options:** (A) Abuse (B) Arrest (C) Arson (D) Stealing

**Output:** <think>During the video, a person is seen entering a room with a bag <time> (30.8, 46.9) </time>. They then proceed to light a fire <time> (170.0, 230.0) </time> using a lighter, which spreads rapidly across the floor. This indicates arson activity. </think>\n<answer>C</answer><glue>[(170.0, 230.0)]</glue>

Figure 6: A qualitative example for long video understanding.

**Question:** Which of the following options correctly sequences the order in which the competition items appear in the video?
**Options:** A. 400m run, shot put, 110m hurdles. B. 100m run, shot put, long jump.

C. 100m run, long jump, shot put. D. 400m run, 110m hurdles, shot put.

**Output:** <think><time>(78.00, 150.00)</time> The video starts with the 100m Decathlon event, where athletes compete in the 100m run. < time >(150.00, 220.00)</time> Next, the Long Jump Decathlon is shown, where athletes perform the long jump. < time >(380.00, 490.00)</time> Finally, the Shot Put Decathlon is displayed, where athletes throw the shot put. This sequence matches the order in the options provided.</think>\n <answer>C</answer>><glue>[(78.00, 150.00), (150.00, 220.00), (380.00, 490.00)]</glue>

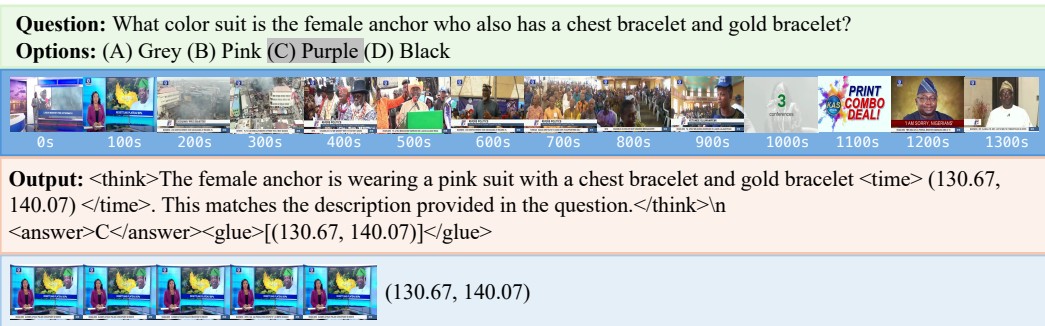

Figure 7: A qualitative example for long video understanding.

**Question:** What color suit is the female anchor who also has a chest bracelet and gold bracelet?
**Options:** (A) Grey (B) Pink (C) Purple (D) Black

**Output:** <think>The female anchor is wearing a pink suit with a chest bracelet and gold bracelet <time> (130.67, 140.07) </time>. This matches the description provided in the question.</think>\n <answer>C</answer><glue>[(130.67, 140.07)]</glue>

(130.67, 140.07)

Figure 8: A qualitative example for long video understanding.

# E    LIMITATION AND FUTURE DIRECTION

Our current approach performs only a single round of zoom-in during both training and inference. We did not explore iterative or recursive zooming due to computational constraints. However, multi-stage zooming could further refine temporal grounding by progressively narrowing the search space and focusing on increasingly fine-grained visual cues.

Another limitation is that the zoom-in process is enforced rather than adaptive. Ideally, the model itself should decide whether, when, and how many times to zoom in, guided by the task objective. A goal-oriented, multi-step zooming policy could potentially yield more efficient and faithful grounding.

**Question:** Why did the woman point at the far end at the end of the video?
**Options:** (A) baby leg is there (B) adjust the camera (C) show him something (D) couple hugging (E) take a photo
**GT Span:** [55.5, 61]

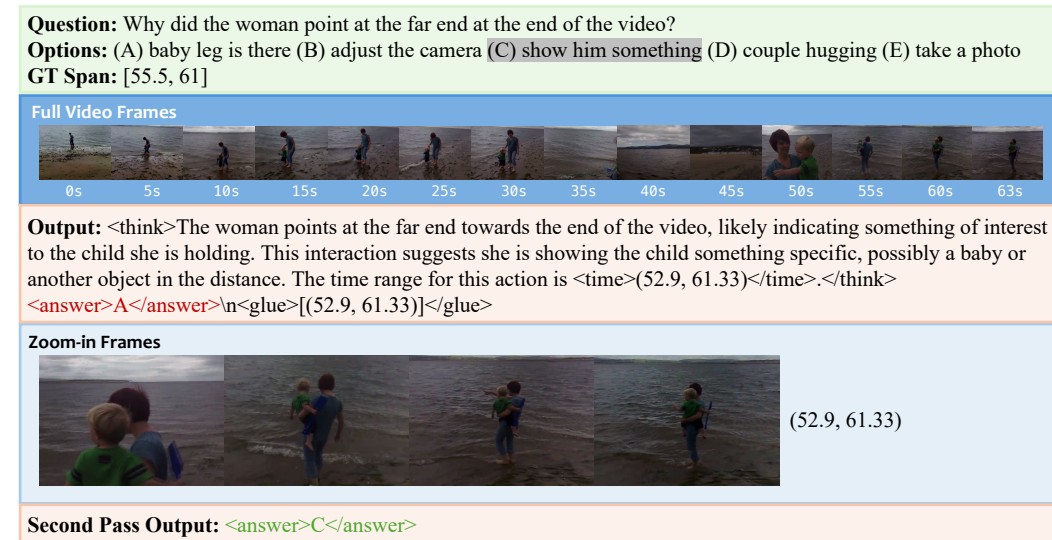

**Output:** <think>The woman points at the far end towards the end of the video, likely indicating something of interest to the child she is holding. This interaction suggests she is showing the child something specific, possibly a baby or another object in the distance. The time range for this action is <time>(52.9, 61.33)</time>.</think>
<answer>A</answer>\n<glue>[(52.9, 61.33)]</glue>

**Zoom-in Frames**

(52.9, 61.33)

**Second Pass Output:** <answer>C</answer>

Figure 9: A qualitative example for grounded videoQA with temporal zoom-in.

**Question:** How many people are at the staircase at the beginning of the video?
**Options:** (A) One (B) Two (C) Three (D) Four

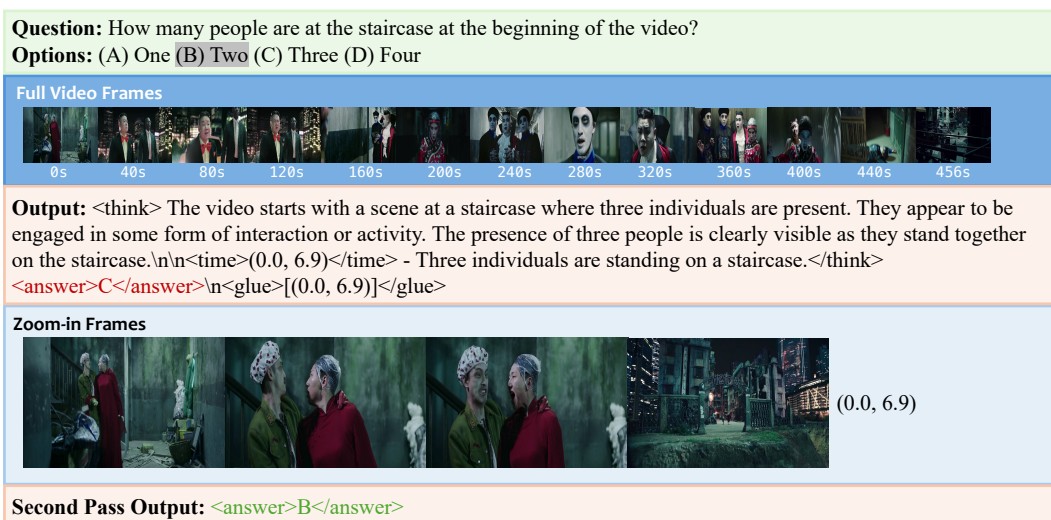

**Output:** <think> The video starts with a scene at a staircase where three individuals are present. They appear to be engaged in some form of interaction or activity. The presence of three people is clearly visible as they stand together on the staircase.\n\n<time>(0.0, 6.9)</time> - Three individuals are standing on a staircase.</think>
<answer>C</answer>\n<glue>[(0.0, 6.9)]</glue>

**Zoom-in Frames**

(0.0, 6.9)

**Second Pass Output:** <answer>B</answer>

Figure 10: A qualitative example for long video understanding with coarse-to-fine zoom-in.

If a strong pretrained model with reasonable temporal grounding ability and exploration samples is sufficient, our framework could also be trained without explicit temporal interval annotations. Instead of relying on rule-based reward, i.e., $R_{\mathrm{IoU}}$, the model could learn to verify whether key visual clues exist within its predicted temporal segments. This self-verification mechanism has the potential to mutually enhance answer accuracy and temporal grounding, especially in long-video scenarios where temporal annotations are often scarce to obtain.

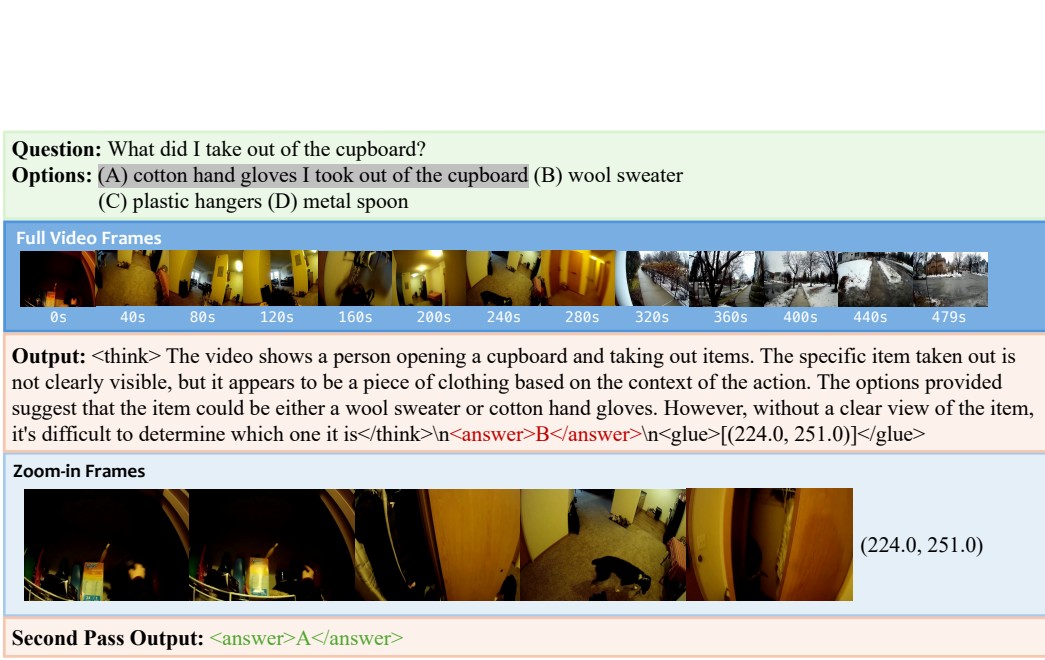

**Question:** What did I take out of the cupboard?
**Options:** (A) cotton hand gloves I took out of the cupboard (B) wool sweater
(C) plastic hangers (D) metal spoon

**Full Video Frames**

0s  40s  80s  120s  160s  200s  240s  280s  320s  360s  400s  440s  479s

**Output:** <think> The video shows a person opening a cupboard and taking out items. The specific item taken out is not clearly visible, but it appears to be a piece of clothing based on the context of the action. The options provided suggest that the item could be either a wool sweater or cotton hand gloves. However, without a clear view of the item, it's difficult to determine which one it is</think>\n<answer>B</answer>\n<glue>[(224.0, 251.0)]</glue>

**Zoom-in Frames**

(224.0, 251.0)

**Second Pass Output:** <answer>A</answer>

Figure 11: A qualitative example for long video understanding with coarse-to-fine zoom-in.

