# OpenReview forum: "Zoom-Zero: Reinforced Coarse-to-Fine Video Understanding via Temporal Zoom-in"
_ICLR.cc/2026/Conference — Submitted to ICLR 2026_

### Official Review · Reviewer_zmbY · 2025-10-15

**Soundness:** 3
**Presentation:** 3
**Contribution:** 3
**Rating:** 6
**Confidence:** 3

**Summary:**

This paper introduces Zoom-Zero, a coarse-to-fine framework designed to improve grounded video question answering (GVQA) by addressing the poor temporal awareness of large video-language models (LVLMs). The method first performs a coarse-grained pass over the entire video to localize query-relevant temporal segments, and then executes a fine-grained pass by "zooming in" on these segments with higher spatial resolution to verify visual evidence. To train this model, the authors enhance the Group Relative Policy Optimization (GRPO) reinforcement learning algorithm with two primary innovations: (1) a zoom-in accuracy reward, which ensures that the localized segments contain the necessary visual information to answer correctly, and (2) a token-selective credit assignment (TokenAdv) mechanism, which decouples multi-faceted rewards and assigns them to the specific tokens responsible for either temporal grounding or answer generation. Extensive experiments demonstrate that Zoom-Zero achieves state-of-the-art performance on several GVQA and long-video understanding benchmarks.

**Strengths:**

* The concept of a coarse-to-fine temporal zoom-in is highly intuitive, mirroring human visual cognition by first getting a general understanding and then focusing on important details. This provides an effective solution to the inherent trade-off between maintaining long-range temporal context and capturing fine-grained visual details.

* The paper clearly identifies a key weakness of the uniform credit assignment problem. The proposed TokenAdv solution, which selectively attributes credit to different parts of the generated text, is a logical and well-justified improvement. Additionally, the zoom-in accuracy reward is a clever mechanism to enforce a stronger link between the localized visual evidence and the final answer.

* The method demonstrates significant performance gains over strong SFT-based and RL-based baselines across multiple challenging benchmarks, including NEXT-GQA, ReXTime, CG-Bench, VideoMME.

**Weaknesses:**

* The two-pass nature of the coarse-to-fine framework inherently increases computational cost and latency at inference time compared to single-pass methods. The "Divide-and-Conquer" strategy, while effective, is even more costly, with a reported 2.3x increase in inference time. A more detailed discussion of this efficiency trade-off would be beneficial.

* The framework's performance is heavily reliant on the initial coarse-grained pass to successfully identify the correct temporal segment. If the initial localization is incorrect, the fine-grained pass has no mechanism for recovery and will analyze an irrelevant portion of the video, potentially leading to a confident but incorrect answer.

**Questions:**

Please see the Weaknesses

---

> ### Author Response · Authors · 2025-11-18
> **Rebuttal by Authors [1/2]**
>
> We sincerely thank you for the valuable suggestion and positive feedback. Below, we address your concerns and provide additional clarifications to strengthen our paper.
>
> ------
>
> >**W1: The discussion of efficiency trade-off of the coarse-to-fine framework.**
>
> Thanks for raising this concern. We provide a clearer breakdown of the effectiveness–latency trade-off in the table below, and report three inference scenarios:
>
> (i) **One-stage inference:** As shown in the second line of the table below, Zoom-Zero uses the same one-stage inference pipeline as the baseline, resulting in nearly identical inference time (it might vary a little due to the number of generated tokens). Trained with our proposed method, this setting yields an average improvement of +1.0 over the baseline without introducing additional latency.
>
> Please kindly note that the main experimental results, as shown in Table1 and Table 2 only have one-stage inference. The zoom-in paradigm is only conducted in Table 4.
>
> (ii) **Two-stage inference (Coarse-to-fine):** The coarse-to-fine variant (third line in the table below) adds a fine-grained pass on grounded frames. This introduces a moderate increase in computation, approximately 1.4× inference time, while delivering a higher average absolute improvement of +2.1 over the baseline.
>
> (iii) **Two-stage inference (Divide-and-conquer):** The divide-and-conquer (fourth line in the table below) scheme is an optional test-time scaling strategy designed to further push performance. While it increases inference time to around 2.3×, it also achieves the largest gain, improving the baseline by +4.3 on average.
>
> | Model | MLVU (Acc) | MLVU (Avg inference time per video) | LVBench (Acc) | LVBench (Avg inference time per video) | VideoMME long w/ sub. (Acc) | VideoMME long (Avg inference time per video) |
> | -------- | :-----: | :-----: | :-----: | :-----: | :-----: | :-----: |
> | Qwen2.5-VL |70.2|18.5s|45.3|39.7s|62.0|25.6s|
> | Zoom-Zero (Ours) |70.8|18.7s|45.7|40.6s|64.2|25.8s|
> | Zoom-Zero + Coarse-to-fine (Ours) |71.4|31.1s|46.3|55.5s|66.2|35.2s|
> | Zoom-Zero + Divide-and-conquer (Ours) |73.4|33.5s|48.1|110.5s|68.7|59.7s|
> | Avg Duration per Video |-|651s|-|4101s|-|2386s|
>
> We also compare with VideoChat-R1.5 [1] (The paper is released on arxiv after our submission) on the test-time scaling setting. Their model is trained on one stage with the same training strategy as VideoChat-R1, but they conduct test-time scaling through multi-round perception during inference (3 rounds by default).
>
> | Model | VideoMME Overall w/o sub. (Acc) | VideoMME (Avg inference time per video) | VideoMME long w/o sub. (Acc) | VideoMME long (Avg inference time per video) | MLVU (Acc) | MLVU (Avg inference time per video) |
> | -------- | :-----: | :-----: | :-----: | :-----: | :-----: | :-----: |
> | VideoChat-R1.5 | 67.1 | 30.0s |57.0|62.4s|70.9|37.1s|
> | Zoom-Zero + Divide-and-conquer (Ours) | 67.9 | 29.8s |57.5|59.7s|73.4|33.5s|
>
> Compared with the test-time scaling method VideoChat-R1.5, our method costs slightly less time while achieving higher long video understanding accuracy.
>
> [1] VideoChat-R1.5: Visual Test-Time Scaling to Reinforce Multimodal Reasoning by Iterative Perception, NeurIPS 2025.

---

> ### Author Response · Authors · 2025-11-18
> **Rebuttal by Authors [2/2]**
>
> >**W2: The framework's performance is heavily reliant on the initial coarse-grained pass to successfully identify the correct temporal segment.**
>
> (i) Tables 1 and 2 report results under the same one-stage inference setting as the baseline, where our model consistently yields higher overall performance. The improvement in mIoU further confirms the enhanced temporal grounding ability of our approach. Moreover, Table 3 shows that our method achieves stronger ground-truth coverage, which verifies the reliability over baselines on temporal zoom-in.
>
> (ii) Table 4 highlights that both the Coarse-to-fine and Divide-and-conquer strategies effectively identify relevant segments and consistently improve long-video understanding performance.
>
> (iii) As we mentioned in line 448, we compute the answer confidence by calculating the probability of the predicted answer token, which also serves as a signal to judge the relevancy of the grounded frames to some extent.
>
> Indeed, temporal localization in long videos remains a challenging task. Our proposed approach that integrates a zoom-in reward into a two-round pipeline, along with a selective credit assignment mechanism, significantly improves grounded QA performance and can be naturally extended to enhance long-video understanding. We hope our proposed method can pave the way for future advances in video temporal reasoning.

---

> > ### Comment · Reviewer_zmbY · 2025-11-19
> >
> > Thanks for the authors' explanations. I have no questions, and I keep my positive assessment for this paper.

---

> > > ### Author Response · Authors · 2025-11-25
> > >
> > > Thank you for taking time to respond to our feedback and appreciate your positive review of our work.

---

### Official Review · Reviewer_Ksr4 · 2025-10-25

**Soundness:** 3
**Presentation:** 4
**Contribution:** 3
**Rating:** 4
**Confidence:** 4

**Summary:**

This paper proposes an improved GRPO algorithm tailored for grounded video question answering by reflecting a temporal zoom-in strategy and a decoupled credit assignment mechanism for multi-task reward optimization. The experiments on NExT-GQA and ReXTime have demonstrated remarkable performance improvements. The method also shows effectiveness for long-form video question answering. Interestingly, the zoom-in idea also works well on a naïve video partition approach.

**Strengths:**

1.	The two innovations -- temporal zoom-in and decoupled credit assignment, sound reasonable and are easy to understand.
2.	The presentation is well-structured and easy to read.
3.	The experiment results are good (yet to be justified, see weakness).

**Weaknesses:**

1.	The coarse-to-fine zoom-in strategy would severely increase the memory and computation burden during GRPO optimization. There is a lack of comparison and analysis on this limitation.
2.	There are several serious problems in Table 1, making the comparison ineffective.
-	First, NExT-GQA only provides temporal labels for validation and test sets. It seems that the authors use the validation set for training and compare with those methods for zero-shot testing in Table 1 (all non-RL methods). This should be explicitly specified for valid comparison.
-	Second, the authors mix up evaluation metrics for QA accuracy (Acc) and Grounded QA Accuracy (Acc@GQA). All compared SFT methods report zero-shot GQA accuracy while Qwen2.5-VL and RL-based methods report QA accuracy without considering grounding. For fair comparison, Acc@GQA is required for all methods; reporting isolated QA and grounding results is less meaningful for grounded VideoQA task.
-	Third, the official Acc of VideoChat-R1 on NExT-GQA is 70.6, but the result given in Table 1 is 69.8. Any explanations for this discrepancy?
3.	The IoG metric is problematic and should be discouraged, as one can obtain perfect IoG value by simply returning the video length as temporal prediction. I recommend to use Intersection over Prediction (IoP) as suggested in NExT-GQA.

**Questions:**

I will increase my score if the weaknesses are addressed in the rebuttal.

---

> ### Author Response · Authors · 2025-11-18
> **Rebuttal by Authors Part [1/2]**
>
> Thank you for your valuable feedback and insightful questions. We will carefully address each concern in detail below.
>
> ------
>
> >**W1: Analysis of computation overhead during GRPO optimization.**
>
> Thank you for raising this important point. Indeed, the fine-grained pass in our coarse-to-fine zoom-in strategy introduces additional computation during GRPO optimization. Below, we clarify how this overhead is mitigated and provide quantitative measurements.
>
> First, the fine-grained pass is executed within the rollout manager, which leverages VLLM and FSDP sharding to efficiently parallelize generation and reduce memory overhead.
>
> In addition, we make efforts to further mitigate the computation cost.
>
> (i) Predictions with no overlap with the ground-truth clue (IoU ≤ 0) should skip the fine-grained pass to reduce cost. However, directly skipping these cases leads to GPU synchronization issues. To address this, we instead feed a dummy video tensor for such cases. As a result, only a subset of samples undergoes the full fine-grained computation, reducing average cost.
>
> (ii) The dominant computation lies in next-token prediction; longer sequences naturally incur higher latency. To reduce this cost, we compute logits using only the hidden state of the final token to obtain the response (one single option letter).
>
> We also report the actual overhead incurred by the zoom-in step. Using 8×A100 GPUs with a global batch size of 64, we measure the average per-step training time and TFLOPs. The temporal zoom-in strategy introduces a 1.15× increase in per-step training time.
>
> | Method | Time per step (mins) | TFLOPs |
> | ---------------- | ------- | ---- |
> | Zoom-Zero w/o zoom-in |13.29 | 551.2 |
> | Zoom-Zero | 15.30 | 585.4 |
>
> Despite adding only a modest computational overhead, our training strategy consistently improves performance across GQA benchmarks, boosting temporal grounding by 5.2% on NExT-GQA and 4.6% on ReXTime, and strengthening long-video understanding with an average overall gain of 1.6%. Moreover, under the temporal zoom-in paradigm, it further pushes performance by an additional 6.4%.
>
> >**W2 (1): The non-RL methods should be explicitly specified in zero-shot testing.**
>
> Thank you for pointing this out. We have updated the PDF to explicitly clarify in the paper that all SFT-based models are evaluated in a zero-shot setting on NExT-GQA.
>
> Regarding RL-based methods such as VideoChat-R1, their paper states in Section 3.3 that they also train on the NExT-GQA validation split, which makes our comparison consistent with their evaluation protocol.
>
> Additionally, for the other two grounded QA benchmarks, ReXTime (Table 1, right) and CG-Bench (Table 2, rightmost), all models are evaluated strictly in the zero-shot setting, ensuring a valid and fair comparison across methods.
>
> >**W2 (2): Acc@GQA is required for Qwen2.5-VL and RL-based methods.**
>
> Thank you for highlighting this issue. We originally reported Acc and mIoU following VideoChat-R1. To better align with the reviewer’s suggestion, we add the Acc@GQA score below, where Acc@GQA measures the percentage of questions that are correctly answered and the IoP is larger than 0.5.
>
> For NExT-GQA, we have updated the table to ensure that all models are evaluated under the same set of metrics, enabling a fair comparison:
>
> | Model | Acc@GQA | mIoU | R@0.3 | R@0.5 |
> | -------- | ------- | ------- | ------- | ------- |
> | Qwen2.5-VL | 18.9 | 20.2 | 31.6 | 18.1 |
> | VideoChat-TPO | 25.5 | 27.7 | 41.2 | 23.4 |
> | TVG-R1 | 22.1 | 29.2 | 41.6 | 20.8|
> | VideoChat-R1| 24.3 | 32.4 | 50.2 | 27.7 |
> | Zoom-Zero (Ours) | **29.0** | **37.6** | **55.6** | **33.8** |
>
> For ReXTime, the test set must be evaluated through the official server, which returns only answer accuracy and IoU-related metrics independently. Therefore, we can only report answer accuracy for this benchmark.
>
> For CG-Bench, we follow the official protocol and report acc.@IoU, which jointly measures answer correctness and temporal grounding quality. This metric appears in Table 2 (rightmost column), and its definition is provided in line 344: a prediction is counted as correct only if the answer is accurate and its IoU exceeds the threshold, averaged over the set {0.1, 0.2, 0.3, 0.4, 0.5}.
>
> We have revised Table 1 and the evaluation metrics section to reflect these changes.

---

> ### Author Response · Authors · 2025-11-18
> **Rebuttal by Authors Part [2/2]**
>
> >**W2 (3): The official Acc of VideoChat-R1 in NExtGQA.**
>
> We carefully reproduced the NExT-GQA results by evaluating the official checkpoint released at OpenGVLab/VideoChat-R1_7B. For video benchmarks, performance is highly sensitive to factors such as the number of video tokens, sampling rate, and the resolution of sampled frames. As described in line 806 Appendix B, we ensure a fair and consistent evaluation setup by fixing the number of video tokens (8192), sampling rate (1fps), and the minimum (16×28×28) and maximum (768×28×28) number of video pixels.
>
> VideoChat-R1 does not provide an evaluation configuration for NExT-GQA. If we strictly follow the default settings in the official evaluation script [evaluate_gqa.py](https://github.com/OpenGVLab/VideoChat-R1/blob/16f03aa4bb470ade41b34bf09272c5466161cc5f/Videochat-R1/src_eval/evaluate_gqa.py#L84), which uses only 3584 video tokens, the reproduced accuracy drops to 67.9, lower than their official result (70.6) and our final reported result (69.8). We have made our best efforts to reproduce the result and make a fair comparison in the same setting. The detailed comparison is shown below.
>
> | Model | Num Video Tokens | Acc | mIoU |
> | ----- | ------- | ------ | ------ |
> | VideoChat-R1 (official in paper) | not available | 70.6 | 32.4 |
> | VideoChat-R1 (Reproduced, code-default) | 3584 | 67.9 | 34.3 |
> | VideoChat-R1 (Reproduced) | 8192 | 69.8 | 32.4 |
> | Zoom-Zero (Ours) | 8192 | **70.7** | **37.6** |
>
> >**W3: The IoG metric is problematic, as one can obtain perfect IoG value by simply returning the video length as temporal prediction. Should consider Intersection over Prediction (IoP) as suggested in NExT-GQA.**
>
> Thank you for the thoughtful concern regarding the IoG metric.
>
> First, we clarify that our temporal grounding evaluation does not rely on IoG. Following standard practice in prior works, we report temporal grounding performance using the widely adopted IoU metric, which directly reflects prediction accuracy.
>
> The usage of IoG is complementary to IoU, in order to illustrate how well the predicted span covers the ground-truth evidence, which is relevant for validating the effectiveness of our temporal zoom-in mechanism rather than assessing grounding precision.
>
> Regarding the concern that one could trivially achieve a perfect IoG by predicting the entire video, in long-video benchmarks such as CG-Bench, the answer-supporting clue occupies only 1.36% of the video. Predicting the whole video would therefore yield extremely low IoU, which is clearly not the behavior exhibited by our model.
>
> Following your suggestion, we also report the IoP metric as shown below:
>
> | Model | NExT-GQA (mIoU) | NExT-GQA (mIoG) | NExT-GQA (mIoP) | ReXTime  val (mIoU) | ReXTime val (mIoG) | ReXTime val (mIoP) |  CG-Bench (mIoU) |  CG-Bench (mIoG) |  CG-Bench (mIoP) |
> | -------- | ------- | ------- | ------- | ------- | ------- | ------- | ------- | ------- | ------- |
> | Qwen2.5-VL |  20.2 | 56.8 | 29.5 | 31.6 | 54.3 | 43.2 | 2.48 | 10.35 | 4.16 |
> |VideoChat-R1| 32.4 | 93.5 | 39.1 | 43.5 | 64.3 | 52.8 |5.91|18.33|7.34|
> |Zoom-Zero (Ours) | **37.6** | **94.7** | **43.2** | **44.7** | **67.6** | **53.5** |**6.68**|**26.15**|**8.25**|
>
> Overall, one should combine all metrics to examine the model’s performance. Our model consistently achieves higher scores across all three metrics. With IoU and IoP constraining overlong predictions, the complementary IoG values further indicate that our method offers better coverage of the ground-truth evidence.

---

> ### Author Response · Authors · 2025-11-25
> **Sincerely looking forward to your post-rebuttal feedback**
>
> As the discussion period draws to a close, we would like to express our sincere gratitude for your time and insightful comments. In our previous response, we carefully reviewed your feedback and provided detailed clarifications on the following points:
>
> + Computational analysis of GRPO training
>
> + Fair comparison practices, including the addition of the Acc@GQA evaluation metric
>
> + Motivation for using IoG, along with additional IoP results
>
> We hope that our responses have fully addressed your concerns. If you have any remaining questions or would like further clarification, please feel free to let us know.
>
> Thank you again for your thoughtful review of our work.

---

> > ### Comment · Reviewer_Ksr4 · 2025-11-26
> >
> > Thanks for the updated results. Most of my concerns are addressed. However, the sharp drop from 70% Acc@QA to 29% Acc@GQA suggests that most correct answers are not well-grounded, which runs counter to the paper’s goal of improving grounded QA. I encourage the authors to discuss more on this discrepancy rather than only updating the results.

---

> > > ### Author Response · Authors · 2025-11-26
> > >
> > > Thank you for raising this concern. We respectfully note, however, that evaluating our method by only examining the drop from Acc to Acc@GQA is not an appropriate judgment criterion. Acc@GQA should be assessed through comparison with baselines, where our method consistently outperforms all prior approaches. We explain the discrepancy below and hope this clarification addresses your concern.
> > >
> > > **Why Acc@GQA is inherently much lower than Acc@QA.**
> > >
> > > Acc@GQA is jointly constrained by both answer correctness and temporal grounding quality. By definition:
> > >
> > > $$\text{Acc@GQA} = \frac{1}{N} \sum_{i=1}^{N} \mathbf{1}\\left(\text{IoP}_i > 0.5 \,\land\, \text{pred}_i = \text{GT}_i\right)$$
> > >
> > > it is upper-bounded by IoP@0.5. Therefore, even with strong QA accuracy, Acc@GQA remains low if temporal localization is challenging. This explains why Acc@GQA is consistently far below Acc@QA not only for our method but also for all existing baselines.
> > >
> > > **The observed gap is a property of the task, not a failure of our method.**
> > >
> > > Below are the Acc@GQA and IoP@0.5 scores for all baselines:
> > >
> > > | Model | Acc@GQA | IoP@0.5 |
> > > | -------- | ------- | ------- |
> > > | Qwen2.5-VL | 18.9 | 25.3 |
> > > | VideoChat-TPO | 25.5 | 32.8 |
> > > | TVG-R1 | 22.1 | 29.6 |
> > > | VideoChat-R1| 24.3 | 34.8 |
> > > | Zoom-Zero (Ours) | **29.0** | **39.6** |
> > >
> > > **IoP does not fully capture saliency grounding.**
> > >
> > > IoP only checks overlap with the annotated segment. When the predicted segment fully covers the salient frames but is slightly longer in duration, it still cannot reach IoP=1.0. Hence, even semantically well-grounded predictions may be penalized.
> > >
> > > **Evidence from prior work confirms this behavior.**
> > >
> > > Grounded QA is indeed a challenging task. For example, the best model (FrozenBiLM) in NExT-GQA (Table 3 in the https://arxiv.org/pdf/2309.01327) achieves **70.8 Acc@QA, but only 17.5 Acc@GQA** and 23.7 IoP@0.5.
> > >
> > > Therefore, the drop from Acc@QA to Acc@GQA should not be interpreted as evidence that our method fails to improve grounding. This drop is inherent to the task and appears across all models. The proper comparison is against the baselines, where our approach consistently achieves superior performance across Acc@GQA, Acc@QA, mIoU, mIoP, and other grounding metrics, demonstrating clear improvements in grounded QA.
> > >
> > > We hope this clarifies why the discrepancy arises, why it is expected for this task, and how our method improves grounded QA over strong baselines.

---

> > > > ### Comment · Reviewer_Ksr4 · 2025-11-26
> > > >
> > > > Thanks for the clarification, and showing the IoP results. I don't have further questions.

---

> ### Author Response · Authors · 2025-11-26
>
> We thank you again for your valuable time and the engaging discussion that helped improve the quality of our paper. We appreciate the updated score and the constructive feedback throughout the rebuttal.

---

### Official Review · Reviewer_u5TQ · 2025-10-31

**Soundness:** 3
**Presentation:** 3
**Contribution:** 2
**Rating:** 4
**Confidence:** 4

**Summary:**

Zoom-Zero proposes a reinforced coarse-to-fine framework for grounded video question answering (GVQA), where the model first localizes query-relevant temporal segments and then “zooms in” for fine-grained visual verification. By introducing a zoom-in accuracy reward and token-selective credit assignment, it enhances evidence faithful temporal grounding and answer accuracy, outperforming prior GRPO-based LVLMs across GVQA and long-video benchmarks.

**Strengths:**

1. **Strong Results.** The proposed model achieves state-of-the-art performance on grounded video QA and long video understanding benchmarks, clearly demonstrating its effectiveness.
2. **Clarity of Writing.** The paper is well-organized and clearly written, allowing readers to easily follow the technical details and rationale of the proposed approach.
3. **Motivation for Token-Level Advantage Estimation.** The paper insightfully identifies and addresses a key limitation in prior GRPO-based methods of collapsing multiple rewards into a single scalar, offering a well-motivated solution through token-level advantage estimation.

**Weaknesses:**

1. **Novelty compared to VideoChat-R1.** The method largely resembles those of VideoChat-R1.
    1. How is the temporal grounding reward different from the IoU reward used in VideoChat-R1?
    2. Similarly, how does the zoom accuracy reward differ from the accuracy or recall reward in VideoChat-R1?
2. **Novelty compared to Qwen2.5-VL + frame selection methods.** Since Qwen2.5-VL inherently supports dynamic frame sampling, the zoom-in capability seems intrinsic to the base model rather than a novel contribution of Zoom-Zero. Therefore, frame selection methods applied to Qwen2.5-VL could achieve similar spatial zoom-in behavior as Zoom-Zero.
3. **Comparison with related works.** Frame selection for long video understanding has been extensively studied [1, 2, 3, 4]. A more comprehensive comparison with these works should be provided.

    [1] Hu et al, M-LLM Based Video Frame Selection for Efficient Video Understanding, CVPR 2025

    [2] Zhang et al., Q-Frame: Query-aware Frame Selection and Multi-Resolution Adaptation for Video-LLMs, ICCV 2025

    [3] Wu et al., AdaFrame: Adaptive Frame Selection for Fast Video Recognition, CVPR 2019

    [4] Tang et al., Adaptive Keyframe Sampling for Long Video Understanding, CVPR 2025

4. **Validation of ‘verifiability’.** The authors claim that the zoom-in accuracy reward “verifies that grounded segments contain requisite evidence to answer the query.” However, no concrete mechanism ensures that the grounded segments indeed contain sufficient information.
    1. How does the presence of answer in zoomed region verified?
    2. What are the actual statistics of zoomed regions that contain versus lack the required evidence?

**Questions:**

Please refer to weaknesses section.

---

> ### Author Response · Authors · 2025-11-18
> **Rebuttal by Authors Part [1/2]**
>
> We sincerely appreciate your time and insightful feedback. We will address your concern and answer your questions below.
>
> ------
>
> >**W1 (1): Novelty compared to VideoChat-R1. How is the temporal grounding reward different from the IoU reward used in VideoChat-R1?**
>
> As we mentioned in line 254, the format (R_{format}), temporal grounding (R_{IoU}), and answer accuracy (R_{Acc}) reward are the basic rewards used in RL-based methods for GVQA tasks, and none of them have been described as the contribution or novelty of our approach.
>
> Instead, we are the first to introduce the temporal zoom-in paradigm to the RL framework, together with a zoom-in reward that verifies the adequacy of the visual evidence from the temporal grounding prediction. We further address the widely discussed issue of uniform credit assignment in RL by proposing a token-selective credit assignment mechanism, which provides fine-grained, reward-specific token-level advantage estimation.
>
> >**W1 (2): How does the zoom accuracy reward differ from the accuracy or recall reward in VideoChat-R1?**
>
> The zoom reward fundamentally differs from the accuracy reward in VideoChat-R1.
>
> This new reward enforces faithful temporal grounding by encouraging the model to explore an accurate interval that, when zoomed into during the fine-grained pass, reliably supports the correct answer. More concretely and as detailed in Section 4.1 and Figure 2, this verification feeds the model the "zoomed-in" frames from the temporal segment it predicted in the coarse pass, processed at a higher spatial resolution. $R_{Zoom}$ is the accuracy of the answer from this pass.
>
> The $R_{Zoom}$ serves as a visual self-verification mechanism to ensure the adequacy of the grounded salient segments. Unlike the standard accuracy reward, $R_{Zoom}$ explicitly detects failures in temporal grounding. If the model predicts a segment that lacks the true visual evidence, the fine-grained “zoom-in” pass on that segment will fail, resulting in $R_{Zoom} = 0$. This mechanism strengthens the faithfulness of visual grounding and improves the consistency between the temporal grounding prediction and the final answer.
>
> >**W1 (3): Novelty compared to Qwen2.5-VL + frame selection methods.**
>
> Thank you for pointing this out. We clarify the differences between our approach and frame-selection methods below.
>
> (i) Unlike traditional frame selection methods [2,4] that rely on CLIP-based similarity and embedding-space searches, our approach leverages the LLM’s intrinsic reasoning capabilities. Our model reasons over complex questions to identify the events that are semantically relevant to the given query.
>
> (ii) Furthermore, our model predicts temporal durations to localize relevant video segments, whereas conventional frame selection methods typically choose a fixed number of frames for an entire benchmark, ignoring the varying distribution of evidence across events. Video reasoning is a dynamic process that should not be limited to a fixed set of frames. Our design promotes flexibility and dynamic reasoning, enhancing the model’s ability to adapt to diverse temporal contexts.
>
> (iii) Most importantly, our method has a learned, reasoning-driven policy to optimize the model. Our novelty lies in incorporating a zoom-in reward into GVQA training, a mechanism that enables the model to learn and optimize evidence-focused behavior during RL post-training rather than relying solely on inference-time heuristics.
>
> We have updated the PDF to incorporate the suggested references in the related work section, clarifying how frame-selection methods differ from our approach.

---

> ### Author Response · Authors · 2025-11-18
> **Rebuttal by Authors Part [2/2]**
>
> >**W2: Comparison with related works of frame selection.**
>
> We thank the reviewer for suggesting a broader comparison with frame selection methods.
>
> As discussed above, temporal grounding or localization is fundamentally different from generic frame selection approaches. We provide a comparison below to contextualize our method. [1] requires training a selector but does not release the model or code. AKS [4] is the *only method with publicly available code* among the suggested references, and it achieves better results compared with [2]. Therefore, we choose AKS and integrate it on Qwen2.5-VL to ensure a fair comparison.
>
> In AKS, the first stage samples video frames at 1 fps, extracts CLIP features, and computes image–text similarity. The selected frames are then fed into the LLM in the second stage for final inference. In the original paper, the number of selected frames is fixed to 64.
>
> Our models can reason over the query and predict the temporal span of relevant events, allowing the duration to vary naturally given different videos. On average, the predicted segments contain about 136 frames for the fine-grained pass. Therefore, we also report AKS results using 136 selected frames as input for its second stage in the table below.
>
> | Model | Stage 1 (Avg frames) | Stage 2 (Avg frames) | VideoMME (Overall w/o sub.) |
> | -------- | ------- | ------- | ------- |
> | Qwen2.5-VL | 64 | - | 63.3 |
> | Qwen2.5-VL | 1fps | - | 65.2 |
> | Zoom-Zero (Ours) | 1fps | - | 66.0 |
> | Qwen2.5-VL + AKS | 1fps | 64 | 64.3 |
> | Qwen2.5-VL + AKS | 1fps | 128 | 64.7 |
> | Qwen2.5-VL + AKS | 1fps | 136 | 65.0 |
> | Zoom-Zero + Coarse-to-fine (Ours)| 1fps | 136 | **67.9** |
>
> >**W3: Validation of ‘verifiability’. (1) How does the presence of answer in zoomed region verified?**
>
> We use two complementary rewards to ensure the presence of visual evidence during training:
>
> (i) Temporal grounding reward $R_{IoU}$. The training annotations include start and end timestamps indicating where the important visual evidence appears in the video. $R_{IoU}$ provides a positive reinforcement signal to responses that are more accurately grounded in these annotated segments, encouraging the model to attend to segments that truly contain the relevant evidence.
>
> (ii) Zoom-in reward $R_{Zoom}$. After the model predicts the start and end timestamps of the relevant segment, it performs a zoom-in self-verification: the selected temporal segments are passed to the model at higher spatial resolution, and the model then predicts the final answer. This mechanism ensures that the answer relies on the visual content within the selected temporal segments.
>
> To avoid any potential misunderstanding, please kindly note that it is better to call it ‘zoomed interval’ instead of ‘zoomed region’ for our method. We actually perform zooming on selected temporal segments, instead of spatial regions. But given the context budget in the LLM, we can enlarge the resolution of those selected temporal segments in the fine-grained pass. We didn’t mention region-level zooming in our paper, and we have noted in line 213 (footnote) for clarification.
>
> >**W4 (2): What are the actual statistics of zoomed regions that contain versus lack the required evidence?**
>
> **Benchmark statistics.** The GVQA benchmarks provide temporal span annotations that indicate the required visual evidence for each query. Figure 9 illustrates an example of how the ground-truth relevant segment (GT Span) looks like.
> + For NExT-GQA, the evidence segments occupy, on average 20.68% of the video.
> + CG-Bench (long GVQA): The answer-supporting segments cover only 1.36% of the entire video on average.
> + General long-video benchmarks (VideoMME, MLVU, LVBench) do not provide start–end annotations for salient segments.
>
> **Model performance.** IoU measures the overlap between the predicted and ground-truth spans divided by their union. IoP (intersection over prediction) measures what fraction of the predicted span contains the required evidence over the whole prediction. IoG complements IoU by measuring how much of the ground-truth span is covered by the prediction.
> We demonstrate our model’s ability to cover the ground-truth segments in Table 3 (we repost the table below), and further illustrate the mIoU across different clue durations and clue proportions in Figure 3 in the Appendix.
>
> | Model | NExT-GQA (mIoU) | NExT-GQA (mIoG) | NExT-GQA (mIoP) | ReXTime  val (mIoU) | ReXTime val (mIoG) | ReXTime val (mIoP) |  CG-Bench (mIoU) |  CG-Bench (mIoG) |  CG-Bench (mIoP) |
> | -------- | ------- | ------- | ------- | ------- | ------- | ------- | ------- | ------- | ------- |
> | Qwen2.5-VL |  20.2 | 56.8 | 29.5 | 31.6 | 54.3 | 43.2 | 2.48 | 10.35 | 4.16 |
> |VideoChat-R1| 32.4 | 93.5 | 39.1 | 43.5 | 64.3 | 52.8 |5.91|18.33|7.34|
> |Zoom-Zero (Ours) | **37.6** | **94.7** | **43.2** | **44.7** | **67.6** | **53.5** |**6.68**|**26.15**|**8.25**|

---

> ### Author Response · Authors · 2025-11-25
> **Sincerely looking forward to your post-rebuttal feedback**
>
> We sincerely appreciate your time and helpful feedback. In our previous reply, we addressed your comments on:
>
> + Clarifying the novelty and explain the fundamental difference with VideoChat-R1 and frame-selection methods
> + Adding quantitative comparisons with frame-selection approaches
> + Explaining verifiability and providing supporting statistics on visual evidence
>
> We hope our clarifications have resolved your concerns. If you have any further questions, please feel free to let us know.
>
> Thank you again for your time and efforts.

---

> > ### Author Response · Authors · 2025-11-27
> >
> > We would like to kindly follow up on our earlier message. We have carefully addressed all of your comments in the rebuttal.
> >
> > We also want to point out that other reviewers who **raised similar concerns now consider their questions resolved** after reading our clarifications. In particular, regarding verifiability, we further demonstrated that IoG complements IoP by capturing the coverage of salient frames, an explanation that Reviewer Ksr4 explicitly acknowledged as sufficient for addressing their concern. They have also indicated a positive inclination toward the paper following this clarification.
> >
> > Please let us know if our responses address your concerns. We sincerely appreciate your time and consideration.

---

### Official Review · Reviewer_WwCb · 2025-11-01

**Soundness:** 3
**Presentation:** 3
**Contribution:** 2
**Rating:** 4
**Confidence:** 4

**Summary:**

The paper tackles a key challenge in Grounded Video Question Answering (GVQA) — ensuring that large video-language models (LVLMs) not only generate correct answers but also ground their reasoning in the correct temporal segments of a video. Zoom-Zero extends the GRPO-based reinforcement fine-tuning framework with two key innovations, including coarse-to-fine temporal zoom-in and zoom-in reward design. Experiments show consistent but moderate improvements over strong baselines, validating the practical value of hierarchical grounding and refined RL training.

**Strengths:**

1. The paper addresses an important and well-defined limitation in current GRPO-based LVLM reinforcement fine-tuning for video understanding — namely, the gap between answer correctness and temporal grounding fidelity. The authors articulate this motivation clearly and convincingly, showing that existing methods often fail to verify whether grounded frames truly contain the visual evidence supporting the answer.
2. The paper does not simply introduce a two-stage pipeline; it integrates it into a reinforcement learning framework in a principled way. The introduction of the Zoom-in Accuracy Reward (RZoom) and Token-Selective Credit Assignment (TokenAdv) demonstrates a nuanced understanding of GRPO’s limitations.
3. The experiments are extensive and well-controlled. The authors benchmark across six datasets (NExT-GQA, ReXTime, CG-Bench, VideoMME, MLVU, LVBench), comparing against both SFT-based and RL-based baselines of similar model scales (7B–8B). The inclusion of both short- and long-video tasks ensures that improvements are consistent across temporal lengths. Results are reported with standard metrics (IoU, mIoU, R@K, accuracy) and are presented clearly.

**Weaknesses:**

1. The novelty of this paper is marginally below the acceptance borderline. While the proposed Zoom-Zero framework introduces a coarse-to-fine “temporal zoom-in” mechanism and two reinforcement learning enhancements (the zoom-in accuracy reward and token-selective credit assignment), these components represent incremental extensions rather than substantial methodological breakthroughs.
2. The overall design is heuristic mainly and system-oriented, building on existing GRPO-based RL fine-tuning frameworks such as VideoChat-R1 and TVG-R1. The “zoom-in” paradigm is conceptually intuitive. It has been explored in prior hierarchical video reasoning works, while the newly introduced rewards and credit assignment strategies refine but do not fundamentally advance the underlying learning algorithm.
3. The paper does not provide a detailed analysis of the computational overhead introduced by the coarse-to-fine zoom-in process. The two-stage inference (coarse localization + fine-grained zoom-in) likely increases latency and resource usage, especially for long videos. This is a practical concern for real-time applications. No comparison of inference time or FLOPs with baseline methods is provided.

**Questions:**

See the weakness part.

---

> ### Author Response · Authors · 2025-11-18
> **Rebuttal by Authors Part [1/3]**
>
> We greatly value your time and feedback. Below, we will carefully address your specific concerns and questions individually.
>
> ------
>
> >**W1: “Temporal zoom-in” mechanism and two reinforcement learning enhancements (the zoom-in accuracy reward and token-selective credit assignment) represent incremental extensions.**
>
> We respectfully disagree with the reviewer’s comment that our contributions are only incremental extensions. By introducing the zoom-in reward and the token-selective credit assignment mechanism (TokenAdv), we provide a substantive improvement to RL-based post-training for GVQA and general video understanding. As shown in Table 5, such design yields consistent gains in both accuracy and temporal grounding over GRPO baselines, indicating that the contribution offers practical and nontrivial value to the task.
>
> Below, we clarify the limitations of prior approaches and explain how our zoom-in reward and TokenAdv improve over them correspondingly.
>
> **On the novelty & contribution of temporal zoom-in and zoom-in reward.**
>
> Our zoom-in paradigm and zoom-in reward are motivated by two fundamental challenges in RL-based long-video training, and they offer concrete benefits:
>
> ***(i) Guiding correct exploration via self-verification.*** Prior GRPO-based methods for GVQA rely solely on IoU as the reward for temporal grounding, such training objectives could not guarantee that localized video segments faithfully contain the visual evidence required for correct reasoning.
>
> We address this by introducing a zoom-in accuracy reward paired with a fine-grained verification pass, which guides the model to correctly explore the temporal context and gather sufficient visual evidence through self-verification. More concretely,
> + The zoom-in reward provides visual verification of whether the grounded segment actually contains the necessary visual evidence for complete reasoning.
> + This self-verification explicitly guides the model toward correct temporal exploration.
> + The mechanism fundamentally advances how RL interacts with multi-stage evidence exploration in long videos.
>
>
> ***(ii) Addressing the context limit.*** RL training is substantially more computationally intensive than SFT because it requires updating a policy model while maintaining a reference model. In this case, we can only manage to train a 7B model with 8k context size. This forces extreme compression of video resolution, making training ineffective because essential visual cues are lost. The zoom-in paradigm resolves this issue by preserving a global context while selectively zooming in on high-resolution, detail-rich segments relevant to the question.
>
> **On the novelty & contribution of token-selective credit assignment.**
>
> The limitation of uniform credit assignment has been widely discussed in RL methods [3,4,5]. Algorithms like GRPO [3] assign the same reward to every token in a sequence based solely on the final outcome, regardless of each token’s actual contribution. StepGRPO [5] discusses this limit and introduces step-level rewards instead of giving one reward for the whole sequence.
>
> [3] DeepSeekMath: Pushing the Limits of Mathematical Reasoning in Open Language Models.
>
> [4] Beyond the 80/20 Rule: High-Entropy Minority Tokens Drive Effective Reinforcement Learning for LLM Reasoning, NeurIPS 2025.
>
> [5] R1-VL: Learning to Reason with Multimodal Large Language Models via Step-wise Group Relative Policy Optimization, ICCV 2025.
>
> This issue is exacerbated in settings with multiple reward factors, where the model should learn which specific tokens contributed to which reward signal. We propose to solve this problem in a more fine-grained way by decoupling the advantage calculation separately for specific tokens. Specifically, the token-level advantage is computed by averaging the relevant task-specific advantages for each token. This design allows the model to attribute feedback at a finer granularity, improving its ability to learn from diverse multi-objective rewards.
>
> This contribution of token-selective credit assignment is not a minor extension to GRPO but represents a fundamental advance for RL-based post-training on the GVQA task. Our token-selective credit assignment obtains substantial performance gains in both accuracy and temporal grounding over the GRPO baselines, as ablations shown in Table 5.

---

> ### Author Response · Authors · 2025-11-18
> **Rebuttal by Authors Part [2/3]**
>
> >**W2 (1): The overall design is heuristic mainly and system-oriented, building on existing GRPO-based RL fine-tuning frameworks such as VideoChat-R1 and TVG-R1.**
>
> The existing GRPO framework does not natively support the multi-step reasoning required by the zoom-in paradigm. VideoChat-R1 is implemented on top of the [open-r1](https://github.com/huggingface/open-r1) framework, and when we attempted to integrate the zoom-in procedure into open-r1, the system could only train with a 4k context window due to insufficient memory-optimization and sharding support.
> Instead, we adopt [Verl](https://github.com/volcengine/verl), which provides more efficient tensor sharding and vLLM-based rollout management. We extend Verl by adding a multi-step zoom-in reasoning pipeline. However, this integration still required substantial engineering effort to address GPU desynchronization issues arising from multi-step rollouts.
>
>
> In addition, to reduce the computational cost of the zoom-in phase, we depart from the standard rollout procedure where the model must generate the full sequence. Instead, we compute the logits directly from the final-token hidden state to obtain the response of the final answer letter.
>
> >**W2 (2): The “zoom-in” paradigm has been explored in prior hierarchical video reasoning works.**
>
> Unlike prior hierarchical video reasoning approaches [1,2], which are predominantly training-free and operate only at inference time, our focus is not on introducing the zoom-in concept itself. Instead, as clarified in our contribution statement (line 108), our novelty lies in incorporating a zoom-in reward into GVQA training, a mechanism that enables the model to learn and optimize evidence-focused behavior during RL post-training rather than relying solely on inference-time heuristics.
>
> To the best of our knowledge, our work is the first to propose an RL-based zoom-in paradigm for multi-step policy learning, enabling the model to learn when and how to perform temporal zoom-in rather than relying on manually designed inference heuristics.
>
> [1] VideoTree: Adaptive Tree-based Video Representation for LLM Reasoning on Long Videos, CVPR 2025.
>
> [2] VideoAgent: Long-form Video Understanding with Large Language Model as Agent, ECCV 2024.

---

> ### Author Response · Authors · 2025-11-18
> **Rebuttal by Authors Part [3/3]**
>
> > **W3: A detailed analysis of the computational overhead introduced by the two-stage inference coarse-to-fine zoom-in process.**
>
> We appreciate raising this concern. We provide a clearer breakdown of the effectiveness–latency trade-off in the table below, and report three inference scenarios:
>
> (i) **One-stage inference:** As shown in the second line of the table below, Zoom-Zero uses the same one-stage inference pipeline as the baseline, resulting in nearly identical inference time (it might vary a little due to the number of generated tokens). Trained with our proposed method, this setting yields an average improvement of +1.0 over the baseline without introducing additional latency.
> Please kindly note that the main experimental results, as shown in Table1 and Table 2, only have one-stage inference. The zoom-in paradigm is only conducted in Table 4.
>
> (ii) **Two-stage inference (Coarse-to-fine):** The coarse-to-fine variant (third line in the table below) adds a fine-grained pass on grounded frames. This introduces a moderate increase in computation, approximately 1.4× inference time, while delivering a higher average absolute improvement of +2.1 over the baseline.
>
> (iii) **Two-stage inference (Divide-and-conquer):** The divide-and-conquer (fourth line in the table below) scheme is an optional test-time scaling strategy designed to further push performance. While it increases inference time to around 2.3×, it also achieves the largest gain, improving the baseline by +4.3 on average.
> | Model | MLVU (Acc) | MLVU (Avg inference time per video) | LVBench (Acc) | LVBench (Avg inference time per video) | VideoMME long w/ sub. (Acc) | VideoMME long (Avg inference time per video) |
> | -------- | :-----: | :-----: | :-----: | :-----: | :-----: | :-----: |
> | Qwen2.5-VL |70.2|18.5s|45.3|39.7s|62.0|25.6s|
> | Zoom-Zero (Ours) |70.8|18.7s|45.7|40.6s|64.2|25.8s|
> | Zoom-Zero + Coarse-to-fine (Ours) |71.4|31.1s|46.3|55.5s|66.2|35.2s|
> | Zoom-Zero + Divide-and-conquer (Ours) |73.4|33.5s|48.1|110.5s|68.7|59.7s|
> | Avg Video Duration |-|651s|-|4101s|-|2386s|
>
> We also compare with VideoChat-R1.5 [3] (The paper is released on arxiv after our submission) on the test-time scaling setting. Their model is trained on one stage with the same training strategy as VideoChat-R1, but they conduct test-time scaling through multi-round perception during inference (3 rounds by default).
>
> | Model | VideoMME Overall w/o sub. (Acc) | VideoMME (Avg inference time per video) | VideoMME long w/o sub. (Acc) | VideoMME long (Avg inference time per video) | MLVU (Acc) | MLVU (Avg inference time per video) |
> | -------- | :-----: | :-----: | :-----: | :-----: | :-----: | :-----: |
> | VideoChat-R1.5 | 67.1 | 30.0s |57.0|62.4s|70.9|37.1s|
> | Zoom-Zero + Divide-and-conquer (Ours) | 67.9 | 29.8s |57.5|59.7s|73.4|33.5s|
>
> Compared with the test-time scaling method VideoChat-R1.5, our method costs slightly less time while achieving higher long video understanding accuracy.
>
> [3] VideoChat-R1.5: Visual Test-Time Scaling to Reinforce Multimodal Reasoning by Iterative Perception, NeurIPS 2025.

---

> > ### Author Response · Authors · 2025-11-25
> > **Sincerely looking forward to your post-rebuttal feedback**
> >
> > We sincerely appreciate your time and valuable feedback. In our earlier response, we addressed your comments on:
> >
> > + Clarifying the novelty and contributions of our work
> >
> > + Providing computational analysis and comparisons with the baseline and TTS method
> >
> > We hope these clarifications have addressed your concerns. Please feel free to reach out if you have any further questions.
> >
> > Thank you again for your review.

---

> ### Author Response · Authors · 2025-11-27
>
> We would like to kindly follow up on our earlier message. We have carefully addressed all of your comments in the rebuttal.
>
> We would also like to note that other reviewers who raised similar concerns now consider their questions resolved after reviewing our clarifications. For example, regarding the efficiency–performance trade-off, we provided a detailed analysis of inference speed on each benchmark alongside the corresponding performance gains. Reviewer zmbY acknowledged this clarification as sufficient for resolving their concern and has since indicated a positive inclination toward the paper.
>
> Please let us know if our responses address your concerns. We sincerely appreciate your time and consideration.

---

### Author Response · Authors · 2025-11-29
**Summary of the rebuttal**

Dear AC, SAC and PC,

We truly appreciate your effort in dedicating extra time and shouldering additional responsibility to ensure fairness throughout the rebuttal. To help ease the burden and provide a quick overview, we kindly summarize the key points of our rebuttal below.

Reviewers acknowledged many merits of our paper, highlighting the clarity of writing, soundness and motivation of the method, and strong empirical results.

+ **Clear writing and presentation:** The paper is “presented clearly”, "well-organized and clearly written", "well-structured and easy to read" and “easy to understand” (@WwCb, @u5TQ,@Ksr4)

+ **Method is insightful and sound:** Reviewers note the paper "clearly identifies a key weakness", "clever mechanism", "sound reasonable", "well-motivated solution", "insightfully identifies and addresses a key limitation", “integrates … in a principled way” “effective solution”. (@u5TQ,@WwCb,@Ksr4,@zmbY)

+ **Empirically strong and well evaluated:** It is described as "logical and well-justified", "experiments are extensive", "benchmark across six datasets", "significant performance gains over strong SFT-based and RL-based baselines". (@u5TQ,@WwCb,@zmbY)

The reviewers’ concerns and questions mainly focused on clarifications and minor additions rather than substantial revisions, which is an encouraging indication that the core approach and empirical results were in solid shape.

**The novelty and contribution (@WwCb, @u5TQ).** We clarify the fundamental difference against previous GRPO-based methods, as well as frame-selection methods. Apart from methodology, we also add quantitative comparison to show the performance superiority against frame-selection methods.

**Computation analysis: training overhead (@Ksr4) and effectiveness–latency trade-off (@WwCb, @zmbY).**
+ **Training.** We make efforts to optimize the framework and further mitigate the computation cost. The temporal zoom-in strategy requires only 1.15× the baseline per-step training time.
+ **Inference.** We provide a detailed breakdown of inference time and performance. With the same one-stage inference, Zoom-Zero improves the baseline by +1.0 with no added latency. Two-stage inference achieves larger gains with moderate overhead: +2.1 absolute gain with 1.4× latency (coarse-to-fine) and +4.3 absolute gain with 2.3× latency (divide-and-conquer). We also compare with very recent work on video test-time scaling, our method is slightly faster while delivering higher long-video understanding accuracy.

**Effective comparison (@Ksr4).**

+ We clarify GVQA comparisons on NExT-GQA and ensure evaluation for all methods in a pure zero-shot setting on ReXTime and CG-Bench.
+ Following the suggestion, we use Acc@GQA to assess both answer correctness and temporal grounding.
+ We clarify that IoG is used not as the primary metric but as a complementary measure to demonstrate coverage, and we have also added IoP results as suggested.

**Verifiability (@u5TQ) and reliability (@zmbY) of temporal zoom-in.**

+ Two rewards R_{IoU} and R_{Zoom} complement each other in verifying the presence of visual evidence during training.
+ We provide benchmark statistics to show the proportion of salience segments and the model performance demonstrating the reliable temporal grounding and coverage ability.
+ Both the Coarse-to-fine and Divide-and-conquer strategies effectively identify relevant segments and consistently improve long-video understanding performance, which also proves the reliability and effectiveness of temporal zoom-in.

In the post-rebuttal period, **Reviewer u5TQ** and **Reviewer WwCb** did not respond before the comment period closed.

**Reviewer zmbY** confirmed that all concerns have been addressed and kept a positive assessment of the paper.

**Reviewer Ksr4** had a follow-up question regarding the gap between Acc@QA and Acc@GQA. We clarified the definition of Acc@GQA, explained why it is naturally lower than Acc@QA, and provided bounded IoP@0.5 along with our improvement over baselines. Then the reviewer was satisfied with our clarification, had no further questions, and raised the score toward acceptance.

We sincerely thank you once again for your time and effort in evaluating our work.

Best regards,

Authors of Paper 7362

---

### Meta-Review · Area_Chair_44vy · 2026-01-09

**Summary:**

This paper proposes a method for doing grounded video QA on long video.  It uses a GRPO-based RL fine-tuning framework.  The proposed contributions are a coarse-to-fine temporal zooming in, and a reward design for RL based on the zooming in.

Reviewers appreciate the strong empirical performance.

Reviewers raise concerns about the
- extra compute requirements of the proposed strategy, especially with respect to the zooming in
- incremental novelty, building on top of very similar existing works that also use GRPO-baesd RL fine-tuning for video QA, VideoChatR1 and TVG-R1; the zooming-in contribution already exists in other works as well
- incorrect usage of dataset splits, incorrect usage of evaluation measures

**Reviewer Concerns:**

The clarification questions were mostly addressed by the rebuttal.  The in-depth issues raised by Ksr4 on the correct usage of dataset splits and evaluation measures are also addressed.  Reviewer Ksr4 was able to raise these concerns because they have in-depth knowledge on NExT-GQA; this does raise concerns on the other evaluations, for which it is possible the reviewers are less knowledgeable.

The main concern which remains unaddressed after rebuttal is the distinction and novelty with respect to existing work.  The authors respond by listing very fine-grained detailed differences which seem more to be small design choice differences.

**Reviewer Scores:**

WwCb - 4; I don't think this reviewer will raise their score, as their raised concern is regarding novelty with respect to existing work, and how differences are incremental
u5TQ - 4; I don' think this reviewer will raise their score, as much of their raised concerns are about novelty with respect to existing methods VideoChat-R1 and the large body of frame-selection works
Ksr4 - 4; this reviewer will likely raise the score to 6 after the rebuttal, as they raised several concerns on the evaluation measures used
zmbY - 6.  I don't think this reviewer would raise their score after rebuttal, as they state that they want to keep their score.

---

### Decision · Program_Chairs · 2026-01-26

Reject